# 3DRot: 3D Rotation Augmentation for RGB-Based 3D Tasks

## Abstract

RGB-based 3D tasks, e.g., 3D detection, depth estimation, 3D keypoint estimation, still suffer from scarce, expensive annotations and a thin augmentation toolbox, since most image transforms, including resize and rotation, disrupt geometric consistency. In this paper, we introduce 3DRot, a plug-and-play augmentation that rotates and mirrors images about the camera's optical center while synchronously updating RGB images, camera intrinsics, object poses, and 3D annotations to preserve projective geometry, achieving geometry-consistent rotations and reflections without relying on any scene depth. We first validate 3DRot on a classical RGB-based 3D task, monocular 3D detection. On SUN RGB-D, inserting 3DRot into a frozen DINO-X + Cube R-CNN pipeline raises $IoU_{3D}$ from 43.21 to 44.51, cuts rotation error (ROT) from 22.91° to 20.93°, and boosts $mAP_{0.5}$ from 35.70 to 38.11; smaller but consistent gains appear on a cross-domain IN10 split. Beyond monocular detection, adding 3DRot on top of the standard BTS augmentation schedule further improves NYU Depth v2 from 0.1783 to 0.1685 in abs-rel (and 0.7472 to 0.7548 in $\delta < 1.25$), and reduces cross-dataset error on SUN RGB-D. On KITTI, applying the same camera-centric rotations in MVX-Net (LiDAR+RGB) raises moderate 3D AP from about 63.85 to 65.16 while remaining compatible with standard 3D augmentations. Because it operates purely through camera-space transforms, 3DRot drops into diverse RGB-based 3D tasks and multi-modal pipelines without architectural changes or depth supervision.

## 1 Introduction

RGB-based 3D perception has become a cornerstone of robotics, autonomous driving, and augmented-reality workflows. However, curating large-scale 3D datasets is far more expensive and time-consuming than labeling 2D images: annotators must specify metric object poses and sizes, often with laser scans or multi-view capture, and reports show that 3D boxes can cost several-fold more per instance than 2D boxes. Data scarcity consequently bottlenecks generalization and fuels the search for synthetic diversity (Brazil et al., 2023). Focusing on non-generative, non-insertion, plug-and-play augmentations, today's RGB-based 3D detection pipelines still rely on a narrow menu-chiefly random scaling and cropping (with intrinsic updates), horizontal flips (and only rarely vertical flips), and global color jitter-because any transform must preserve geometric consistency so that augmented objects remain physically plausible in the scene (Brazil & Liu, 2019; Chen et al., 2020; Li et al., 2022; 2024; Zou et al., 2021). Indeed, naively pasting objects at arbitrary depths or orientations injects implausible scale-depth cues and can hurt detector accuracy (Ge et al., 2023; Li et al., 2024; Parihar et al., 2025).

Beyond the ubiquitous color jitters trick inherited from 2D detection pipelines, 3D perception currently relies on three fairly disjoint families of data-augmentation techniques:

**(1) Geometry-Consistent Camera/Scene Transforms.** These methods perturb the camera pose or the ground plane and re-project all annotations, thereby preserving the exact 2D–3D correspondence. Typical operations include scaling, cropping, ground–plane-constrained rotations, and flipping (Engilberge et al., 2023; Lian et al., 2022). In multi-sensor fusion and multi-view settings, flipping is a common choice as long as the transformation is applied consistently across modalities/views (Wang et al., 2021; Pan et al., 2025). The advantages are simplicity, no rendering pipeline, and plug-and-play integration; however, the operation set remains narrow, and rotation augmentation is restricted

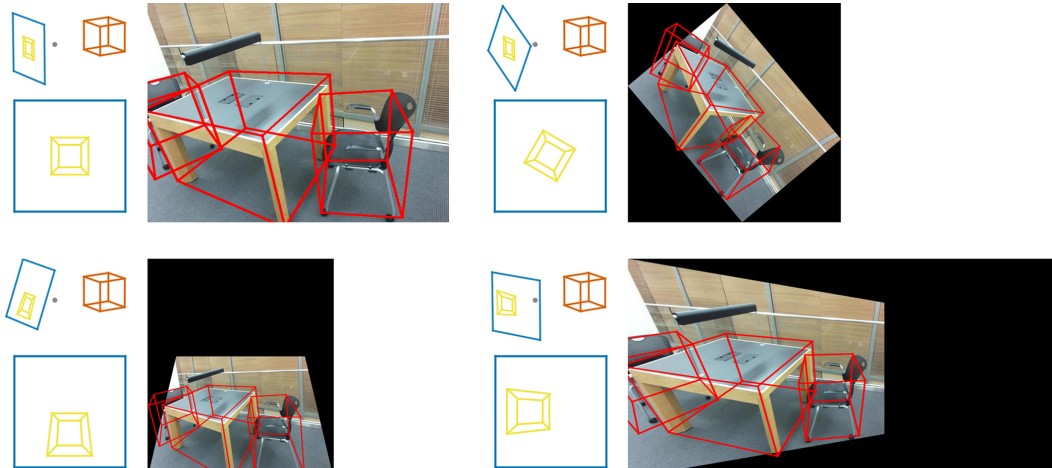

Figure 1: Overall concept of 3DRot. We rotate images about the camera's optical center and synchronously update intrinsics, poses, and labels to preserve projective geometry. In each panel, the left subfigure is a concept sketch: red denotes the 3D bounding box, blue the screen border, and yellow the projected 3D box on the screen. Panels: top-left (origin), top-right (yaw 50°), bottom-left (roll 20°), bottom-right (pitch 20°).

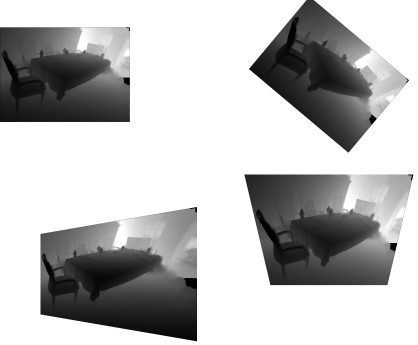

Figure 2: Depth maps. The block is a $2 \times 2$ grid shown in row-major order with camera rotations about the optical center: $(0°, 0°, 0°)$, yaw $+40°$, pitch $+20°$, and roll $+20°$. In all cases 3DRot applies the same pure-rotation homography to the RGB image and updates labels/intrinsics accordingly, so the depth remain 2D–3D consistent while the image footprint changes. See Supplementary Video for an animated demo.

to in-plane/coplanar assumptions and thus cannot be directly used when views or data are not approximately coplanar. **(2) Physically Plausible Instance/Object Insertion.** These approaches cut object instances from real scenes or CAD repositories, paste them in 3D or BEV, and synchronize the RGB projections while explicitly handling occlusion and scale (Ge et al., 2023; Parihar et al., 2025; Wang et al., 2021). The main drawback is the heavy overhead of scene reconstruction and photometric harmonization (e.g., collision checks, lighting/shadow consistency), which hampers scalability across domains. **(3) Synthetic / Generative and Weak-Supervision Pipelines.** These pipelines first reconstruct the 3D scene (or novel viewpoints), then edit objects or cameras in the 3D world and re-render to produce training images (Wang et al., 2024; Cheng et al., 2025; Chang et al., 2024). Geometric consistency is guaranteed by the rendering process and the resulting realism can be photo-realistic; nevertheless, reconstruction and rendering are computationally expensive, and iteration cycles are slow.

Recent progress has gravitated toward physically realistic instance insertion and generative/weakly supervised pipelines rather than toward truly plug-and-play primitives. Object insertion methods synthesize diversity by pasting CAD or real instances with explicit occlusion/scale handling (Ge

et al., 2023; Parihar et al., 2025), while generative routes reconstruct scenes or viewpoints and then re-render (Wang et al., 2024; Cheng et al., 2025; Chang et al., 2024). Consequently, the most fundamental RGB-based setting, monocular RGB, still relies primarily on horizontal flips and mild color jitter (Brazil & Liu, 2019; Li et al., 2022; Zou et al., 2021; Chen et al., 2020; Li et al., 2024), while in-plane rotation-an indispensable and highly effective augmentation in 2D recognition-remains largely untapped in 3D settings. Even the ubiquitous horizontal flip still lacks a principled account of when and why it helps. These observations motivate us to devise a plug-and-play, geometry-aware augmentation module that enriches object-pose diversity without information loss, alleviates annotation scarcity, and lays the groundwork for more sophisticated, geometry-driven augmentations in future work.

In this paper, we propose Optical Center 3D Rotation Augmentation (3DRot), a plug-and-play module that enriches RGB-based 3D training data by rotating and mirroring scenes about the camera's optical center while rigorously preserving projective geometry as shown in Fig. 1. For each desired in-plane rotation or reflection, 3DRot derives an exact homography from the camera projection matrix, enabling pixel remapping with no need to simulate occlusions, shadows, or lighting; the same closed-form transformation keeps the RGB image, 3D data, and camera intrinsics mutually consistent, delivering depth-free, geometry-faithful augmentation that drops seamlessly into existing 3D tasks. Moreover, multi-modal signals (e.g., LiDAR point clouds and depth-sensor maps) can be updated in lockstep with the RGB using the same camera-centric rotation/reflection, addressing the long-standing issue of cross-modal augmentation asynchrony in 3D detection (Wang et al., 2021; Zhang et al., 2020) ; see Fig. 4 and 2 for depth-map and NOCS (Krishnan et al., 2024) examples. The formulation is especially pertinent for platforms with rapidly changing orientations—e.g., UAVs, aerospace imaging payloads, and dynamic robots—where robustness to roll/pitch/yaw is crucial. We validate 3DRot on monocular 3D cuboid detection, monocular depth estimation, and LiDAR+RGB detection, demonstrating that it is task- and modality-agnostic.

As a fundamental RGB-based 3D task, monocular 3D detection serves as our main testbed. We insert 3DRot into a frozen DINO-X (Ren et al., 2024; Liu et al., 2024) + Cube R-CNN head (Brazil et al., 2023) with matched schedules. On the SUN RGB-D 10-category split (SUN10) (Song et al., 2015), 3DRot improves $IoU_{3D}$ from 43.21 to 44.51, lowers ROT from 22.91° to 20.93°, and increases $mAP_{0.5}$ from 35.70 to 38.11; smaller but consistent gains appear on a cross-domain IN10 split. Ablations confirm geometry-consistent rotations and chirality-safe flips as the main drivers, with principal-point realignment providing a minor additional benefit.

Beyond monocular detection, we show that the same camera-centric formulation extends to both dense and multi-modal 3D tasks. Plugging 3DRot into the BTS monocular depth estimator yields additional gains on top of its standard augmentation recipe, reducing NYU Depth v2 abs-rel from 0.1783 to 0.1685 and improving $\delta < 1.25$ from 0.7472 to 0.7548, while also lowering cross-dataset error on SUN RGB-D. On KITTI, applying 3DRot to MVX-Net (LiDAR+RGB) improves moderate 3D AP from about 63.85 to 65.16 and remains compatible with GlobalRotScaleTrans and other standard 3D augmentations. Together, these results position 3DRot as a simple, depth-free primitive that can be dropped into diverse RGB-based 3D pipelines with minimal changes.

## 2 RELATED WORK

Compared with 2D vision benchmarks, 3D datasets are markedly smaller, offer fewer augmentation strategies, and therefore generalize less well. Yet real-world 3D perception must cope with dramatic viewpoint changes: robot wrist joints and pan-tilt heads constantly pitch and yaw; drones undergo rapid yaw and roll in flight, tilting the camera's optical axis; dashcams, roadside infrastructure cameras, and vehicle-mounted sensors experience pitch/roll disturbances during braking or on inclines; and every hand-held AR/MR capture introduces random orientations with each lift or wrist twist. Robust, geometry-consistent augmentations are thus essential for closing the gap between limited training data and these demanding, off-axis scenarios.

### 2.1 RGB-BASED ROTATION AUGMENTATION

Early attempts to regularize RGB-based networks with viewpoint changes started from settings in which the depth channel is readily available, so the augmentation can be expressed directly in camera

space. Keypoint-based Monocular 3D Object Detector (Kim et al., 2020) injects small perturbations into the robot-mounted camera's roll, pitch and yaw and then re-project SUN RGB-D boxes to match the new pose. While this improves robustness to head motions, it still relies on full RGB-D frames. Two-level Data Augmentation (Engilberge et al., 2023) pushes consistency a step further: their two-level scheme first applies an in-plane homography to each calibrated view and then corrects the per-view projection matrix so that all images still meet at a common ground plane. The operation is, however, intrinsically *coplanar*; any 3D structure that rises above or falls below the reference plane (e.g. shelves or hanging objects) accumulates geometric error after transformation. Most recently, GroundMix (Meier et al., 2024) shows that simple in-plane rotations around the optical axis suffice for monocular car- and drone-view detectors; they adjust 3D yaw accordingly and keep the remaining pose angles fixed. This assumption holds only when the camera truly spins about its $z$-axis; moreover, it offers no rigorous proof of geometric consistency, merely demonstrating visual plausibility.

## 2.2 RGB-BASED FLIPPING AUGMENTATION

Random horizontal flipping is the most widely used geometry-aware transform in RGB-based tasks because it is *label-preserving* for the image plane and inexpensive to implement. The pioneering (Brazil & Liu, 2019) pipeline reflects each 3D box corner on the image, then back-projects the flipped vertices with the original depth to obtain an updated 3D cuboid, leaving all other pose parameters. Although effective for heading predictions, this strategy cannot propagate sign changes to the off-diagonal entries of a full $3\times3$ rotation matrix and therefore assumes that only the yaw angle is modeled. Several follow-up methods (Chen et al., 2020; Zou et al., 2021; Li et al., 2022; 2024; Wang et al., 2021; Pan et al., 2025) inherit the same augmentation. Cube R-CNN (Brazil et al., 2023) introduces a more principled update: it composes two mirror matrices, one for the image warp and one for the camera coordinate frame, so that the predicted $R \in SO(3)$ remains valid after a flip. This design finally enables horizontal mirroring for full-angle pose regression, but the derivation presumes that the principal point coincides with the image center; when intrinsics are asymmetric or cropped, the 2D-3D alignment still drifts.

## 2.3 OTHER AUGMENTATIONS

Beyond rotations and flips, several geometry–aware strategies have been explored to enrich RGB-based 3D data. Geometric Consistency Augmentation (Lian et al., 2022) introduces four camera-centric manipulations-random scaling, random cropping, camera translation, and instance-level copy-paste-each designed to preserve depth cues after image warping. While these augmentations improve robustness, random cropping can discard large portions of background context, and the copy-paste pipeline requires dense depth maps plus heuristic rules to blend pasted objects seamlessly. 3D Copy-Paste (Ge et al., 2023) makes this idea more rigorous by sampling physically plausible insertions from an external CAD library: candidate objects are resized, posed, and lit to avoid collisions and match the scene illumination, yielding more realistic composites. Nevertheless, its rendering loop and collision search add notable pre-processing cost, and downstream accuracy still depends on perfect plane reconstruction and light estimation. Moreover, random cropping risks discarding the background context that conveys important geometric cues, while instance-level copy paste-despite its popularity-relies on intricate manual placement heuristics and still falls short of producing truly photorealistic composites (Li et al., 2024).

## 3 METHOD

3DRot applies an arbitrary camera-centric rotation (pitch, roll, yaw; optional mirroring) about the optical center and warps the image using the corresponding pure-rotation homography. This keeps updates to the camera frame and screen space consistent and preserves the projection geometry. We first outline the framework and notation, then derive the rotation model and label-synchronization rules. Intuitively, 3DRot rotates each pixel's viewing ray about the optical center, preserving all geometric correspondences without introducing any new quantities that would need to be re-estimated. A conceptual illustration of this camera-centric rotation and synchronized label update is shown in Fig. 3.

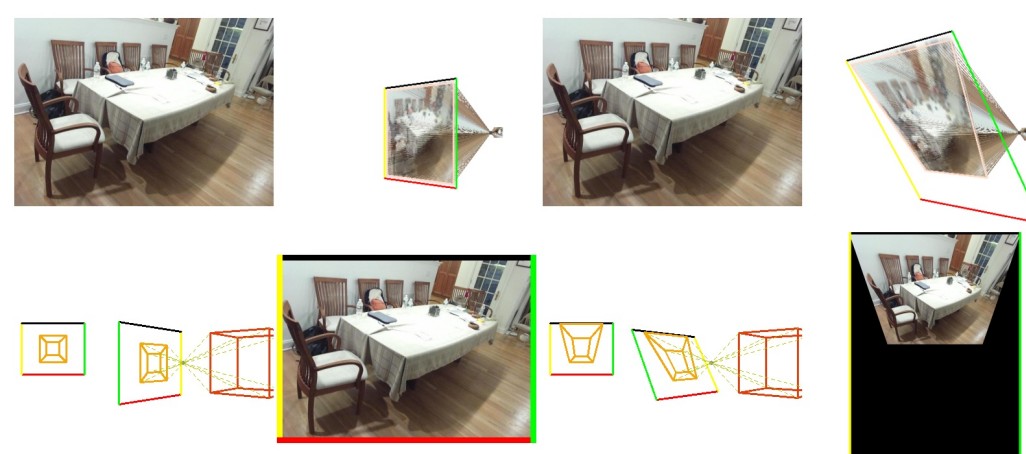

Figure 3: Left: original image; right: view rotated about the optical center with roll $+30°$. Insets: the ray-imaging diagram (top-right) rotates the image plane about the optical center while keeping the viewing rays fixed, and the pose-imaging diagram (bottom-left) rotates the camera frame so that 3D labels remain valid. The resulting RGB on the right preserves 2D–3D geometric consistency while only the screen footprint changes. See Supplementary Video for an animated demo.

To maintain the geometric projection relationship, two principal transformations must be rigorously defended. Firstly, the transformation defining the object's pose relative to the camera must remain consistent with the applied rotation. Secondly, the projective transformation from the camera coordinate system to the image plane must be preserved.

## 3.1 COORDINATE SYSTEM AND 3D CUBOID TRANSFORMATION

3DRot is implemented as a rigid rotation of the camera coordinate frame about the optical center, followed by re-expressing all scene geometry in the rotated camera frame. In the standard world-to-camera formulation, this amounts to composing the original camera extrinsics with the sampled rotation. For individual 3D points, their coordinates in the camera frame are simply left-multiplied by the sampled camera rotation. For axis-aligned cuboids parameterized by a rotation, side lengths, and a 3D center, the same camera rotation is applied to the cuboid rotation and center, while leaving the side lengths unchanged. Appendix B provides the full derivation starting from the world-to-camera transform and shows how this point-wise update lifts to the cuboid representation used throughout this paper.

## 3.2 PROJECTIVE TRANSFORMATION

Having established how the 3D coordinates of each scene point are updated under a camera-centric rotation, we now turn to their image-plane projections. Specifically, we must determine how every pixel location transforms so that the rotated view remains geometry-consistent with the new camera frame. To this end, we (i) recap the pinhole-projection equation, (ii) derive the homography that arises when the camera rotates about its optical center, and (iii) show how this homography simplifies to a pure-rotation form that is valid for *any* 3D scene, without the usual planarity assumption or depth information. These steps yield a closed-form mapping $H = K'R_cK^{-1}$ which we will use to warp the RGB image and masks in the remainder of the pipeline.

The projective transformation from the camera coordinate system to the image plane can be represented by the following.

$$P = \frac{1}{z}KT_c \tag{1}$$

where $P = [u, v, 1] \in \mathbb{R}^{3\times1}$ represents a pixel location in the image, $z$ represents the depth of the 3D point in camera space, and $K$ represents camera's intrinsics.

Equation (1) defines the projection for a single point. If we consider a scenario where the observed 3D points all reside on a single plane $S$, the relationship between $A$ camera view and $B$ camera view of this plane can be simplified. In such cases, any two images of the same planar surface in space are related by a homography:

$$P_A = \frac{z_B}{z_A} K_A H_{AB} K_B^{-1} P_B \tag{2}$$

where $P_A \in \mathbb{R}^{3\times 1}$ belongs to image of camera A, $P_B \in \mathbb{R}^{3\times 1}$ belongs to image of camera B. $H_{AB} \in \mathbb{R}^{3\times 3}$ is defined by:

$$H_{AB} = R_{AB} - \frac{t_{AB}\, n^T}{d} \tag{3}$$

where $R_{AB} \in \mathbb{R}^{3\times 3}$ is defined by $T_A = R_{AB} T_B$, $t_{AB} \in \mathbb{R}^{3\times 1}$ is defined by $T_A = T_B + t_{AB}$. $d \in \mathbb{R}^1$ represents the distance from the optical center of camera $B$ to plane $S$, $n^T \in \mathbb{R}^{3\times 1}$ represents the normal vector of the single plane $S$ and $n^T S + d = 0$. (Hartley & Zisserman, 2004; Szeliski, 2022)

The application of Eq. (3) for the homography $H_{AB}$ presupposes that the observed 3D points lie on a common plane, denoted as $S$. This coplanarity constraint is embedded within the definition of $H_{AB}$.

However, in the specific context of rotational data augmentation, the camera undergoes rotation around its optical center without any translational movement. Consequently, the translation vector $t_{AB}$ becomes zero ($t_{AB} = 0$). Substituting this into Eq. (3), the homography matrix simplifies to $H_{AB} = R_{AB}$.

Therefore, for data augmentation involving only rotation around the optical center, Eq. 2 can be reformulated as:

$$P_A = \frac{z_B}{z_A} K_A R_{AB} K_B^{-1} P_B \tag{4}$$

Crucially, this simplified Eq. (4) no longer requires the coplanarity of 3D points. As a result, Eq. (4) is applicable to any arbitrary 3D scene, provided that the camera transformation consists solely of rotation about its optical center. Hence we treat the transformation as a pure rotation about the optical center, omitting any translation term.

It is important to note that $P_A$ and $P_B$, while elements of $\mathbb{R}^{3\times 1}$, represent points on their respective image planes in homogeneous coordinates, typically of the form $[u, v, 1]^T$. Given that the third component of these homogeneous image coordinates is inherently 1 (after normalization), the scaling factor $\frac{z_B}{z_A}$ in Eq. (4) can be conceptualized as a normalization constant denoted by $\lambda$. This scalar $\lambda$ ensures that the resulting vector $P_A$ is correctly scaled to its homogeneous representation where the last component is 1. Thus, Eq. (4) can be expressed in a more general form, highlighting the projective transformation up to a scale factor:

$$P_A = \lambda\, K_A R_{AB} K_B^{-1} P_B \tag{5}$$

Here, $\lambda$ encapsulates the necessary scaling (equivalent to $\frac{z_B}{z_A}$) to maintain the equality for the chosen homogeneous representation of $P_A$. This form underscores that the geometric core transformation that relates the image points under pure camera rotation is $K_A R_{AB} K_B^{-1}$, with $\lambda$ accounting for the projective scaling.

By synthesizing Eq. (9), Eq. (1), and Eq. (5), we can articulate the comprehensive transformation from a world coordinate system to its projection on the image plane after camera rotation:

$$P_A = \lambda\, K_A\, (R_{AB} R_B\, T_w + R_{AB} t_B) \tag{6}$$

### 3.3 Flipping Transformation

Although analogous to rotation, flipping (reflection) is geometrically distinct because it reverses chirality. In our framework we implement flips as linear operators $M$ acting on the camera frame and object pose, and show—both via a projection-matrix view and via a direct camera-space construction—that pure flips and flip–rotation compositions $R_c = M R$ preserve 2D–3D geometric consistency when applied consistently to poses and projections. Our formula directly achieves a chiral reflection in camera space: we first reflect the image around the optical center, then re-orthogonalize the camera basis (Gram–Schmidt) and enforce a right-handed frame by negating the

third camera basis vector (the first encodes gravity, the second horizontal orientation, and the third is free and used to fix chirality), so the final rotation remains in $SO(3)$ and avoids chirality-flipping ambiguities during projection. The detailed derivations of how $M$ transforms 3D points, poses, and image coordinates, how to compose $M$ with a preceding rotation, and how these viewpoints lead to consistent image-plane projections are deferred to Appendix C.

## 3.4 GENERAL LINEAR TRANSFORMATION

The derivation in Eq. (14) does not hinge on $M$ being a pure reflection; any invertible linear operator $A \in \mathbb{R}^{3 \times 3}$ can replace $M$. The camera-to-image mapping remains a single matrix product $H_A = K_A A K_A^{-1}$, so 2D-3D points stay geometry-consistent under the new view. However, unless $A$ is a scaled rotation, an axis-aligned cuboid does not remain a cuboid under $A$, so the $(R, t, s)$ parameterization cannot be preserved for arbitrary object orientations. We give a short proof in Appendix D.

## 3.5 IMAGE PADDING AND PRINCIPAL-POINT REALIGNMENT

Rotating an image about the pitch or roll axes warps its footprint on the image plane; the resulting view no longer fits the original rectangular support. A naive crop or resize would either discard valid pixels or break the projective consistency implied by the intrinsic matrix. We render the rotated view on a minimal bounding canvas whose center is set to the updated principal point, so that all valid pixels are kept and the intrinsics remain geometry-consistent. The three-step construction is detailed in Appendix E.

## 4 EXPERIMENTS

### 4.1 DATASETS AND METRICS

We evaluate 3DRot on three representative RGB-based 3D settings: monocular 3D cuboid detection, monocular depth estimation, and LiDAR+RGB 3D detection.

**SUN10 and IN10 (monocular 3D detection).** Following OmniNOCS (Krishnan et al., 2024), we adopt its SUN RGB-D slice (Song et al., 2015) and retain the ten furniture-centric categories *chair, table, desk, sofa, bed, nightstand, bookcase, dresser, toilet, bathtub*. To assess cross-domain generalization, we further curate an IN10 split that merges SUN RGB-D, ARKitScenes (Baruch et al., 2021) and Hypersim (Roberts et al., 2021) using the same ten categories. Following Cube R-CNN, we discard objects with invalid depth or marginal image overlap and fix boxes whose third rotation-matrix column is not aligned with gravity. We report the seven metrics used in NOCS-Former and OmniNOCS (Krishnan et al., 2024): $IoU_{3D}$, TRANS [cm], ROT [deg], SIZE [cm], 5GRAV, 10HEAD, and $mAP_{0.5}$. $IoU_{3D}$ is computed using the exact volumetric intersection between predicted and ground-truth cuboids (rather than the fast approximation in the original Cube R-CNN code); AP denotes mean Average Precision at 2D IoU 0.5, and $mAP_{0.5}$ averages AP over the nine 3D IoU thresholds $\{0.05, 0.10, \ldots, 0.50\}$. 5GRAV is the percentage of predictions whose gravity axis is within $5°$ of the ground truth, and 10HEAD is the percentage whose heading (yaw) error is below $10°$.

**NYU Depth v2 and SUN RGB-D (monocular depth).** For depth estimation we insert 3DRot into a BTS ResNet-50 model (Lee et al., 2021) and follow its training protocol on the official NYU Depth v2 split (Nathan Silberman & Fergus, 2012). We evaluate in-domain on the NYU Depth v2 test set and cross-dataset on the SUN RGB-D test images, using the standard single-image depth metrics from Eigen et al. (Eigen et al., 2014); we adopt their original definitions (see Appendix F).

**KITTI (LiDAR+RGB 3D detection).** To study compatibility with LiDAR+RGB pipelines and standard 3D augmentations, we plug 3DRot into MVX-Net (Sindagi et al., 2019) on the KITTI 3D object detection benchmark (Geiger et al., 2013). We follow the official protocol and use the object-detection split with categories *Car*, *Pedestrian*, and *Cyclist*, reporting Average Precision (AP) for 3D boxes ($AP_{3D}$), bird's-eye-view boxes ($AP_{BEV}$), 2D boxes, and orientation similarity (AOS) at

| Method | $IoU_{3D}\uparrow$ | Trans (cm)$\downarrow$ | Rot (deg)$\downarrow$ | Size (cm)$\downarrow$ | 5GRAV$\uparrow$ | 10HEAD$\uparrow$ | $mAP_{0.5}\uparrow$ |
|---|---|---|---|---|---|---|---|
| SUN10 | 43.21 | 11.60 | 22.91 | 12.93 | 78.03 | 47.32 | 35.70 |
| SUN10 + 3DRot | **44.51** | **11.19** | **20.93** | **12.45** | **79.89** | **55.53** | **38.11** |
| IN10 | 30.39 | 22.97 | 31.34 | 13.62 | 79.71 | **38.01** | – |
| IN10 + 3DRot | **30.59** | **22.75** | **31.23** | **13.57** | **81.20** | 37.80 | – |

Table 1: Single-run results on SUN10 and IN10. All models share the same frozen DINO-X + Cube R-CNN backbone and training schedule. 3DRot applies a camera-centric rotation (with probability 0.8; yaw $\pm 10°$, pitch/roll $\pm 5°$) plus a chirality-preserving horizontal flip with probability 0.5; rotation and flip are composable.

| Method | $IoU_{3D}\uparrow$ | Trans (cm)$\downarrow$ | Rot (deg)$\downarrow$ | Size (cm)$\downarrow$ | 5GRAV$\uparrow$ | 10HEAD$\uparrow$ | $mAP_{0.5}\uparrow$ |
|---|---|---|---|---|---|---|---|
| BS (Baseline) | 43.21 | 11.60 | 22.91 | 12.93 | 78.03 | 47.32 | 35.70 |
| BS + 0.5 F | 43.35 | 11.24 | 25.36 | 12.68 | 26.59 | 47.97 | 35.84 |
| BS + 0.5 F(kc) | 43.79 | 11.43 | **20.91** | 12.61 | 78.67 | 55.23 | 36.58 |
| BS + 0.8 $R_{10-5-5}$ | 44.43 | **11.10** | 21.16 | 12.79 | 78.94 | 54.17 | **38.29** |
| BS + 0.8 $R_{30-5-5}$ | 43.89 | 11.22 | 21.70 | 12.60 | **81.30** | 52.38 | 37.53 |
| BS + 0.8 $R_{10-5-5}$ + 0.5F(kc) | **44.51** | 11.19 | 20.93 | **12.45** | 79.89 | **55.53** | 38.11 |

Table 2: Ablation on SUN10. All runs share the same schedule as Table 1.

the official IoU thresholds (0.7 for Car, 0.5 for Pedestrian and Cyclist) and *moderate* difficulty.[1] Our ablations focus on the moderate $AP_{3D}$ scores, which are the most commonly reported indicator of overall 3D performance on KITTI.

## 4.2 IMPLEMENTATION DETAILS

We briefly describe how we integrate 3DRot into each backbone; full architecture, optimization, and hyperparameter details are deferred to Appendix G.

**Monocular 3D detection (SUN10/IN10).** For SUN10 and IN10 we attach a Cube R-CNN head (Brazil et al., 2023) to a frozen DINO-X detector (Ren et al., 2024; Liu et al., 2024), which we use purely as an off-the-shelf 2D object recognizer. 3DRot is applied during training as a camera-centric rotation (optionally combined with a chirality-preserving horizontal flip) that is consistently propagated to the RGB image, camera intrinsics, and 3D cuboid annotations.

**Monocular depth (NYU Depth v2 / SUN RGB-D).** For depth estimation we use the BTS ResNet-50 implementation from Monocular-Depth-Estimation-Toolbox (Li, 2022) and follow its default training schedule on NYU Depth v2. 3DRot is inserted as an additional geometric augmentation in the image pre-processing stage, where the same camera-centric transform is applied to the RGB image, intrinsics, and depth map. All BTS results are from our reimplementation using the official released code, rather than copied from the original BTS paper.

**LiDAR+RGB 3D detection (KITTI / MVX-Net).** For LiDAR+RGB 3D detection on KITTI we adopt the MVX-Faster-RCNN configuration in MMDetection3D (Contributors, 2020), which combines a 2D image backbone with a voxelized LiDAR branch. 3DRot is implemented as an extra camera-centric operator that jointly transforms the RGB image, camera intrinsics, and projected LiDAR points and is composed with the standard GlobalRotScaleTrans and RandomFlip3D augmentations. All MVX-Net results are from our MMDetection3D reimplementation with the official configuration, not directly copied from the original paper.

## 4.3 MAIN RESULTS

**Monocular 3D detection (SUN10 / IN10)** Table 1 summarizes single-run performance on the SUN10 and IN10 splits. On SUN10, inserting 3DRot into a frozen DINO-X + Cube R-CNN pipeline improves all seven pose metrics: $IoU_{3D}$ increases from 43.21 to 44.51, ROT decreases from 22.91° to 20.93°, and $mAP_{0.5}$ rises from 35.70 to 38.11, with smaller but consistent gains on the cross-domain IN10 split. Table 3a further shows that, even when trained only on SUN RGB-D, our

---

[1] KITTI switched from 11 to 40 recall positions ($AP_{R40}$) in 2019; our experiments follow the current official setting.

| Method | Train set | $IoU_{3D}$ ↑ |
|--------|-----------|--------------|
| Cube R-CNN | SUN RGB-D | 36.2 |
| Cube R-CNN | OMNI3DIN | 37.8 |
| **Ours** | SUN RGB-D | 43.21 |
| **Ours + 3DRot** | SUN RGB-D | **44.51** |

| Method | Center | Chir. | $IoU_{3D}$ ↑ | ROT↓ | $mAP_{0.5}$ ↑ |
|--------|--------|-------|--------------|------|---------------|
| Baseline | – | – | 43.70 | 22.22 | – |
| 3DRot (full) | √ | √ | **44.58** | **20.78** | **38.06** |
| w/o Center | × | √ | 44.43 | 20.81 | **38.06** |
| w/o Chirality | √ | × | 44.40 | 23.57 | 37.49 |

(a) Evaluated on the SUN RGB-D 10-category split, using a category set closely aligned with Cube R-CNN.

(b) Ablation on SUN10. We remove objects shorter than 6.25% of image height to avoid downsampling loss. *Center*: image padding & principal-point realignment (without it, we simply crop to the rotated footprint). *Chir.*: chirality preservation during flips.

Table 3: Overall comparison (left) and ablations (right).

| Method | NYU abs_rel ↓ | NYU rmse ↓ | NYU a1 ↑ | SUN abs_rel ↓ | SUN rmse ↓ | SUN a1 ↑ |
|--------|---------------|------------|----------|---------------|------------|----------|
| Baseline | 0.1783 | 0.5683 | 0.7472 | 0.2502 | 0.6738 | 0.6049 |
| 3DRot | **0.1685** | **0.5670** | **0.7548** | **0.2333** | **0.6612** | **0.6120** |

Table 4: Effect of inserting 3DRot into the BTS ResNet-50 training schedule. Both models are trained on NYU Depth v2 and evaluated on NYU Depth v2 and SUN RGB-D; the Baseline uses the default BTS augmentation recipe (including standard 2D rotation and horizontal flips), while 3DRot adds our camera-centric 3D rotation on top of this schedule.

detector with 3DRot reaches 44.51 $IoU_{3D}$, outperforming Cube R-CNN trained on either SUN RGB-D (36.2) or OMNI3DIN (37.8).

**Monocular depth estimation (NYU Depth v2 / SUN RGB-D).** For depth estimation, we plug 3DRot into the BTS ResNet-50 model trained on NYU Depth v2. As shown in Table 5, within the projective-geometry family 3DRot outperforms both horizontal flipping and 2D in-plane rotation on NYU and on cross-dataset SUN RGB-D. Compared to the baseline BTS schedule (Table 4), our final 3DRot configuration further reduces NYU abs-rel from 0.1783 to 0.1685 and improves $\delta < 1.25$ from 0.7472 to 0.7548, while also lowering abs-rel and rmse on SUN RGB-D.

**LiDAR+RGB 3D detection (KITTI).** For LiDAR+RGB 3D detection on KITTI, we insert 3DRot into MVX-Net within MMDetection3D. Table 6a shows that a small camera-centric yaw+pitch configuration ($R_{3-3-0}$) improves moderate Overall $AP_{3D}$ from 63.85 to 65.16 without degrading Car AP, whereas a roll-only setting ($R_{0-0-3}$) noticeably hurts Cyclist and Overall AP, which is consistent with KITTI's acquisition setup where the cameras are mounted approximately level with the ground plane, so large synthetic roll angles produce atypical viewpoints. Table 6b further indicates that 3DRot can be combined with standard scene-level augmentations such as GlobalRotScaleTrans and RandomFlip3D, with the best configuration including 3DRot remaining within a few AP points of the strongest non-3DRot setting.

Additional quantitative results and full tables are deferred to Appendix H.

### 4.4 ABLATION STUDY

**Ablations on monocular 3D detection.** Table 2 and Table 3b dissect the SUN10 gains. Naively adding a horizontal flip (BS + 0.5F) slightly improves $IoU_{3D}$ but severely harms pose quality (ROT rises to 25.36° and 5GRAV collapses), showing that mirroring a full 3×3 rotation matrix without chirality control is unsafe. Enforcing chirality preservation (BS + 0.5F(kc)) restores low rotation error and yields a substantial 10HEAD gain, while pure camera-centric rotations ($R_{10-5-5}$, $R_{30-5-5}$) improve most metrics and combining them with chirality-safe flips gives the best overall trade-off. Table 3b further shows that padding and principal-point realignment bring a small but consistent benefit, whereas disabling chirality preservation significantly degrades orientation metrics, confirming that the improvements come from geometry-consistent rotations and chirality-safe reflections rather than generic image warping.

**Ablations on monocular depth estimation.** For depth estimation, Table 5 isolates projective augmentations within a fixed BTS training recipe. Within this projective family, 3DRot consistently outperforms horizontal flips and 2D in-plane rotations on both NYU Depth v2 and SUN RGB-D, indicating that explicitly updating intrinsics and camera-space rays is more effective than treating rotations as purely 2D warps. Comparing Table 4 and Table 5 further shows that these gains

| Family | Aug. | NYU | | | SUN | | |
|---|---|---|---|---|---|---|---|
| | | abs ↓ | rmse ↓ | a1 ↑ | abs ↓ | rmse ↓ | a1 ↑ |
| base | plain | 0.2105 | 0.6765 | 0.6582 | 0.2891 | 0.7587 | 0.5366 |
| proj_family | flip | 0.2027 | 0.6828 | 0.6574 | 0.2704 | **0.7408** | 0.5485 |
| proj_family | 2DRot | 0.2059 | 0.6460 | 0.6818 | 0.2835 | 0.7567 | 0.5343 |
| proj_family | 3DRot | **0.1940** | **0.6370** | **0.6954** | **0.2684** | 0.7414 | **0.5494** |
| color_family | color | **0.1879** | **0.6102** | **0.7134** | **0.2532** | **0.6867** | **0.5932** |

Table 5: BTS ResNet-50 trained on NYU Depth v2 with different augmentation families, evaluated on NYU (left) and SUN RGB-D (right). `proj_family` applies projective augmentations to RGB and depth jointly: a horizontal flip, an in-plane 2D rotation baseline (2DRot, treating depth as a fourth image channel but without enforcing a true roll around the principal point), or our camera-centric 3DRot. `color_family` adds stronger photometric jitter. Projective and color augmentations can be combined in the full schedule; here we isolate them to highlight their individual effects.

| Exp. | Overall | Ped. | Cyc. | Car |
|---|---|---|---|---|
| BASE | 63.85 | 55.12 | 58.13 | **78.31** |
| $R_{3-0-0}$ | 62.97 | 56.25 | 56.12 | 76.55 |
| $R_{0-3-0}$ | 64.87 | 57.78 | **59.32** | 77.52 |
| $R_{0-0-3}$ | 58.58 | 54.98 | 45.56 | 75.21 |
| $R_{3-3-0}$ | **65.16** | **59.26** | 57.94 | 78.27 |

| $(G, F, R)$ | 3D box | BEV | AOS |
|---|---|---|---|
| $(0, 0, 0)$ | 44.78 | 53.91 | 50.08 |
| $(1, 0, 0)$ | 61.86 | 68.96 | **66.69** |
| $(0, 1, 0)$ | 51.00 | 59.06 | 54.61 |
| $(0, 0, 1)$ | 50.80 | 59.65 | 54.58 |
| $(1, 1, 0)$ | **63.62** | **70.53** | 66.63 |
| $(0, 1, 1)$ | 56.45 | 63.47 | 62.59 |
| $(1, 0, 1)$ | 61.18 | 68.85 | 64.55 |

(a) Moderate 3D AP on KITTI for Overall and each class under different AugRotate3D settings $R_{a-b-c}$, where $a$, $b$, and $c$ denote the maximum yaw, pitch, and roll angles (in degrees) used by AugRotate3D.

(b) Effect of configurations $(G, F, R)$ for GlobalRotScaleTrans, RandomFlip3D, and AugRotate3D.

Table 6: MVX-Net ablations on KITTI: (left) impact of AugRotate3D on class-wise moderate 3D AP; (right) interaction between GlobalRotScaleTrans (G), RandomFlip3D (F), and AugRotate3D (R), where each configuration is denoted as $(g, f, r) \in \{0, 1\}^3$. GlobalRotScaleTrans is a pure point-cloud augmentation and does not modify the 2D images.

are additive to standard color-based augmentation: even with strong photometric jitter, inserting a small-angle 3DRot still reduces abs-rel and improves $\delta < 1.25$ on both datasets.

**Interaction with standard 3D augmentations in MVX-Net.** We also examine how 3DRot interacts with the standard 3D scene-level augmentations in MVX-Net, namely GlobalRotScaleTrans and RandomFlip3D (Table 6b). Turning on GlobalRotScaleTrans alone $(1, 0, 0)$ yields a large improvement over the no-augmentation setting $(0, 0, 0)$, raising moderate 3D AP from 44.78 to 61.86 and AOS from 50.08 to 66.69, while adding RandomFlip3D $(1, 1, 0)$ gives the highest 3D AP of 63.62. Configurations that also include 3DRot $(0, 0, 1), (0, 1, 1), (1, 0, 1)$ obtain 3D AP between 50.80 and 61.18 with AOS in the mid-50s to mid-60s; in particular, the best 3DRot configuration $(1, 0, 1)$ is within about 2.5 points of $(1, 1, 0)$ in 3D AP and roughly 2 points in AOS. Overall, these patterns suggest that camera-centric rotations can be combined with existing 3D augmentations in MVX-Net without destabilizing detection performance, while leaving headroom for further tuning of angle ranges and scheduling.

## 5 CONCLUSION & DISCUSSION

We have introduced 3DRot, a simple yet effective rotation-and-reflection augmentation that operates directly at the camera optical center. 3DRot does not require depth information for scene reconstruction while maintaining 2D-3D geometric consistency. Despite these benefits, 3DRot has limitations. It is most effective when camera poses are diverse; on KITTI, where cameras are roughly level with the ground plane, strong roll perturbations can hurt performance (Table 6a). When intrinsics are missing they must be estimated from 2D–3D correspondences, and large intrinsic variation across a dataset may also require a normalization step, e.g., mapping all views to a shared "virtual camera" as in Cube R-CNN (Brazil et al., 2023). We discuss in Appendix I.

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

## A    ADDITIONAL DEPTH AND NOCS VISUALIZATIONS

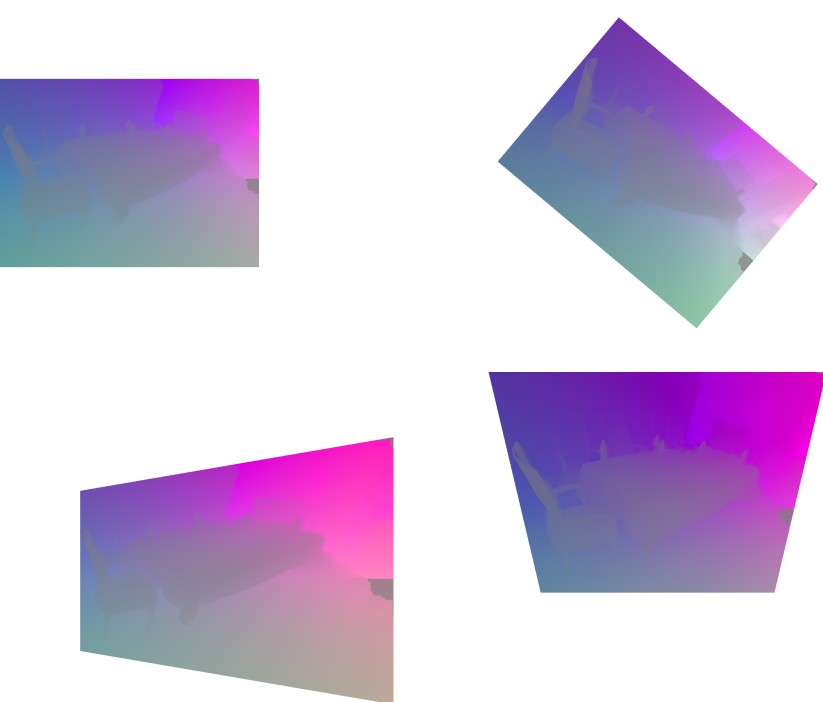

Figure 4: NOCS (R/G/B encode X/Y/Z) maps under camera-centric rotations about the optical center. The block is a $2 \times 2$ grid shown in row-major order with camera rotations: $(0°, 0°, 0°)$, yaw $+40°$, pitch $+20°$, and roll $+20°$. In all cases 3DRot applies the same pure-rotation homography to the RGB image and updates labels/intrinsics accordingly, so the depth and NOCS remain 2D–3D consistent while the image footprint changes.

## B  CAMERA-CENTRIC ROTATION OF 3D POINTS AND CUBOIDS

To propagate a virtual camera rotation to all scene geometry, we first clarify how 3D points migrate between the world frame, the original camera frame, and the rotated camera frame. This section revisits the world-to-camera transform, derives the point-wise update induced by a camera-centric rotation, and then lifts the result to the cuboid pose used in the main paper.

A point expressed in the world frame is mapped to the camera frame through

$$T_c = R T_w + t, \tag{7}$$

where $T_c \in \mathbb{R}^{3 \times 1}$ denotes a 3D point in the camera coordinate system, $T_w \in \mathbb{R}^{3 \times 1}$ denotes the same point in the world coordinate system, and $R \in \mathbb{R}^{3 \times 3}$ and $t \in \mathbb{R}^{3 \times 1}$ are the camera extrinsics.

After rotating the camera by a camera-centric rotation matrix $R_c$, the new camera coordinate system is

$$T_{\text{rot-c}} = R_c T_c. \tag{8}$$

Combining Eq. 7 and Eq. 8 gives

$$T_{\text{rot-c}} = R_c R T_w + R_c t, \tag{9}$$

which shows that applying a camera-centric rotation requires updating the camera extrinsics from $(R, t)$ to $(R_c R, R_c t)$.

The derivation above describes how any single 3D point is transferred from the original camera frame to the rotated one. We now extend this result to an entire axis-aligned cuboid whose pose is parameterized by its rotation, side lengths, and translation. Let

$$T_{bb} = R_{bb} S_{bb} + t_{bb} \tag{10}$$

denote the $3 \times 8$ matrix of the eight cuboid corners in the *original* camera coordinates, where the column matrix $S_{bb} \in \mathbb{R}^{3 \times 8}$ is formed by concatenating the vectorized offsets $\left( \pm \frac{W}{2}, \pm \frac{H}{2}, \pm \frac{L}{2} \right)^{\top}$.

Applying the camera-centric rotation in Eq. equation 8 to every corner gives

$$\begin{aligned} T_{bb}^{\text{new}} &= R_c \big( R_{bb} S_{bb} + t_{bb} \big) \\ &= \underbrace{(R_c R_{bb})}_{R_{\text{new}}} S_{bb} + \underbrace{R_c t_{bb}}_{t_{\text{new}}}. \end{aligned} \tag{11}$$

Hence the updated pose parameters are

$$R_{\text{new}} = R_c R_{bb}, \qquad S_{\text{new}} = S_{bb}, \qquad t_{\text{new}} = R_c t_{bb}. \tag{12}$$

That is, the cuboid's orientation and translation are left-multiplied by the camera rotation $R_c$, whereas its side lengths remain unchanged. This update rule is used throughout our experiments when applying 3DRot to 3D bounding boxes.

## C  FLIPPING TRANSFORMATION DETAILS

Although analogous to rotation, flipping (reflection) augmentation is geometrically distinct. Rotations preserve chirality, while reflections invert it. Consequently, the geometric consistency of the reflection transformation must be validated for its use in our framework.

Accordingly, this subsection (i) formalizes a reflection with a matrix $M$, (ii) derives how $M$ transforms 3D points and their image projections, and (iii) shows how to compose $M$ with a preceding rotation so that the combined operator $R_c = M R$ can be applied in a single step. These results guarantee that flipping, alone or in combination with a rotation, preserves the geometric relationships required by our augmentation pipeline.

Let the projection in the original camera space be defined by Eq. (13):

$$z_A P_A = K_A T_A \tag{13}$$

After a reflection defined by a transformation matrix $M$, a 3D point $T_A$ in the original camera space is mapped to a new point $T_{AM}$ in the mirrored space:

$$T_{AM} = M T_A \tag{14}$$

The projection of this new point is subsequently governed by Eq. (15):

$$z_{AM} P_{AM} = K_{AM} T_{AM} \tag{15}$$

where $K_{AM}$ represents the intrinsic parameters corresponding to the mirrored view.

By substituting the object pose from Eq. (7) and Eq. (13) into Eq. (14), we can determine the new pose in the mirrored camera system. The original pose $(R_A, t_A)$ is transformed into $(M R_A, M t_A)$. Consequently, the comprehensive transformation from the world's coordinate system to the mirrored image plane is given by:

$$P_{AM} = \lambda K_{AM} (M R_A T_w + M t_A) \tag{16}$$

This demonstrates that reflection can be consistently integrated as a valid geometric transformation applied directly to the object's pose parameters.

Furthermore, this formulation allows reflection to be seamlessly composed with rotation. A composite transformation can be applied where the object's pose $(R_A, t_A)$ is first rotated by $R$ and subsequently reflected by $M$. The final pose becomes $(M R R_A, M R t_A)$, leading to the final projection equation:

$$P_{AM} = \lambda K_{AM} (M R R_A T_w + M R t_A), P_{MR} = \lambda K_{MR} M R K_A^{-1} P_A, H_M = K_{MR} M R K_A^{-1}$$

This confirms the geometric consistency of combining these augmentation techniques.

# D  Loss of Cuboid Structure Under General Affine Maps

**Loss of cuboid structure.**  Unlike reflections or rotations, a general linear (or affine) transform distorts the unit axes unequally; therefore an axis-aligned 3D bounding box is mapped to a *parallelepiped* rather than a rectangular cuboid. To retain the canonical $(R, t, s)$ representation after an affine transform $A \in GL(3)$, we require that the transformed cuboid again factorizes into an *orthogonal* rotation, a *diagonal* size matrix, and a translated center. Applying $A$ to the eight corners of the original box yields with $R_A \in SO(3)$ and $D_A = diag(d_1, d_2, d_3)$ yet to be determined.

$$\mathbf{T}_{bb}^{new} = A(R_{bb}S_{bb} + t_{bb}) = \underbrace{(R_A R_{bb})}_{R_{new}} \underbrace{(D_A S_{bb})}_{S_{new}} + \underbrace{A\,t_{bb}}_{t_{new}},$$

The required factorization can exist only if:

$$AR_{bb} \;=\; R_A R_{bb} D_A$$

We require an affine matrix $A \in \mathbb{R}^{3 \times 3}$ such that, for *every* orthogonal matrix $O_1 \in O(3)$, there exist an orthogonal matrix $O_2$ and a diagonal matrix $D$ satisfying

$$A\,O_1 \;=\; O_2\,D.$$

Multiplying on the left by $O_1^\top$ and on the right by its transpose gives

$$O_1^\top A^\top A\,O_1 \;=\; D^\top O_2^\top O_2\,D \;=\; D^2.$$

Because the scene may contain objects under *arbitrary* orientations, the relation must hold for every rotation.

$$O_1^\top A^\top A\,O_1 \;=\; D^2, \quad \forall O_1 \in O(3)$$

Setting $O_1 = I$ yields $A^\top A = D^2$. Now recall the uniqueness theorem of second-order isotropic tensors (Reddy, 2018) : If a real symmetric tensor T satisfies $Q\,T\,Q^\top = T$ for all $Q \in O(3)$, then $T = \lambda I$ .

Applying this to the symmetric tensor $T = A^\top A$ gives $A^\top A = \lambda^2 I$, hence $A = \lambda R_A$ for some $\lambda > 0$ and $R_A \in O(3)$. In the context of axis-aligned cuboids we further fix $R_A = I$, so the only admissible affine map is a uniform scaling $A = \lambda I$. Affine transforms with shears or non-uniform scalings therefore cannot preserve the $(R, t, s)$ factorization for all poses and are left to the *Discussion* section for domain-specific treatment.

## E   IMAGE PADDING AND PRINCIPAL-POINT REALIGNMENT

We prevent this with a three-step procedure that both preserves every pixel and keeps the principal point at the center of the final canvas:

1. **Project the original corners.** Let $P_{\text{orig}}$ be the four image corners and $R$ the applied camera rotation. Define $K_0$ to equal the source intrinsics $K_A$ but with principal point $(0, 0)$. The corners in the rotated frame are

$$P'_{\text{rot}} \propto K_0 \, R \, K_A^{-1} \, P_{\text{orig}}.$$

2. **Compute the bounding box.** After homogeneous normalization, write the image-space coordinates as $(u'_j, v'_j)$. The half-width and half-height of the minimal axis-aligned box centered on the optical axis are

$$x_{\max} = \max_j |u'_j|, \qquad y_{\max} = \max_j |v'_j|.$$

3. **Update intrinsics and crop.** Build a new intrinsic matrix $K_C$ equal to the source intrinsics $K_A$ but with principal point $(x_{\max}, y_{\max})$. Render the rotated image with $K_C$ and finally crop the rectangle $[0, 2x_{\max}] \times [0, 2y_{\max}]$.

The $2x_{\max} \times 2y_{\max}$ canvas now contains every valid pixel from the rotation, while its center $(x_{\max}, y_{\max})$ exactly coincides with the principal point of $K_C$, preserving internal geometry.

# F  DEPTH METRICS

For completeness, we recall the standard single-image depth metrics from Eigen et al. (Eigen et al., 2014). Let $d_i$ and $\hat{d}_i$ denote the ground-truth and predicted depth at pixel $i$, and $N$ the number of valid pixels. We report absolute relative error (abs-rel), root mean squared error (rmse), and the $\delta < 1.25$ accuracy (a1), defined as

$$\text{abs-rel} = \frac{1}{N} \sum_i \frac{|d_i - \hat{d}_i|}{d_i}$$

$$\text{rmse} = \sqrt{\frac{1}{N} \sum_i (d_i - \hat{d}_i)^2}$$

$$\delta < 1.25 = \frac{1}{N} \left| \{i : \max(d_i/\hat{d}_i, \hat{d}_i/d_i) < 1.25\} \right|$$

# G   IMPLEMENTATION DETAILS

## G.1   MONOCULAR 3D DETECTION (SUN10/IN10)

The last decoder layer produces $N{=}900$ object queries (256 dims). We feed the corresponding image features into a Cube R-CNN head (Brazil et al., 2023), which predicts a 9-D cuboid representation-2D center, depth, 3D dimensions, and a full $3{\times}3$ rotation matrix.
We train for $1\,000$ epochs using AdamW (lr $= 2{\times}10^{-4}$), MultiStepLR ($\gamma = 0.9$ at every $100$ epochs), and clip gradients to an $L_2$ norm of 35.

## G.2   MONOCULAR DEPTH ESTIMATION (NYU DEPTH V2 / SUN RGB-D)

For depth estimation we use the BTS ResNet-50 model from Monocular-Depth-Estimation-Toolbox, implemented as a `DepthEncoderDecoder` with a `BTSHead` and the standard scale-invariant loss. On NYU Depth v2 we train for 60 epochs with AdamW (learning rate $1{\times}10^{-4}$) and a OneCycle learning-rate schedule, using a batch size of 64 and $416{\times}544$ crops after the NYU-specific `NYUCrop`. 3DRot is realized as an `AugRotate3D` operator inserted after `NYUCrop` and before the usual `RandomFlip`, `RandomCrop`, and `ColorAug`: with probability 0.7 it applies a small camera-centric rotation (yaw/pitch up to $3°$, roll up to $5°$) and updates the RGB image, depth map, and camera intrinsics consistently. All other data-processing steps and evaluation metrics follow the official NYU Depth v2 configuration.

## G.3   LIDAR+RGB 3D DETECTION (KITTI / MVX-NET)

For LiDAR+RGB detection we follow the official MVX-Faster-RCNN configuration in MMDetection3D, using a `DynamicMVXFasterRCNN` detector with a ResNet-50–FPN image backbone, a DynamicVFE + SparseEncoder LiDAR branch, and an Anchor3DHead. We train on the KITTI three-class benchmark for 50 epochs with AdamW (learning rate $3{\times}10^{-3}$), a cosine-annealing schedule with linear warm-up, and a batch size of 8, starting from the released pre-trained MVX-Faster-RCNN checkpoint. The baseline pipeline includes `GlobalRotScaleTrans` and `RandomFlip3D` as standard 3D scene-level augmentations. 3DRot is implemented as an `AugRotate3D` operator placed before these two transforms: with probability 0.3 it samples a camera-centric rotation with yaw and pitch in $\pm3°$ (zero roll) and applies it jointly to the RGB image, calibration matrices, LiDAR points, and 3D boxes. All other hyperparameters and evaluation metrics are kept identical to the default MVX-Net KITTI setup.

# H  ADDITIONAL QUANTITATIVE RESULTS

| Method | IoU$_{3D}\uparrow$ | Trans (cm)$\downarrow$ | Rot (deg)$\downarrow$ | Size (cm)$\downarrow$ | 5GRAV$\uparrow$ | 10HEAD$\uparrow$ |
|---|---|---|---|---|---|---|
| BASE | 42.43 | 11.87 | 23.99 | 13.09 | 77.03 | 45.39 |
| BASE + Color jitter | 41.51 | 12.24 | 22.48 | 13.07 | **78.84** | 48.16 |
| BASE + 3DRot | **44.41** | **11.13** | **21.91** | **12.50** | 77.26 | **49.06** |

Table 7: Validation results with different augmentations. Trans and Size are reported in centimeters; Rot is the mean absolute rotation error in degrees. 5GRAV denotes accuracy within $5°$ of the ground-truth gravity axis, and 10HEAD denotes accuracy within $10°$ of the ground-truth heading. All configurations are trained without any flipping augmentation.

| Aug method | NYU abs_rel$\downarrow$ | NYU rmse$\downarrow$ | NYU a1$\uparrow$ | SUN abs_rel$\downarrow$ | SUN rmse$\downarrow$ | SUN a1$\uparrow$ |
|---|---|---|---|---|---|---|
| No AUG | 0.1841 | 0.6088 | 0.7089 | 0.2449 | 0.6676 | 0.6065 |
| 2DROT 2.5 (baseline) | 0.1783 | 0.5683 | 0.74722 | 0.2502 | 0.6738 | 0.6049 |
| 2DROT 5 | 0.1775 | 0.5707 | 0.74602 | 0.2477 | 0.6717 | 0.6047 |
| 2DROT 10 | 0.1729 | 0.5704 | 0.74793 | 0.2460 | 0.6776 | 0.5996 |
| 3DROT 1-1-2.5 | 0.1711 | 0.5679 | 0.75202 | 0.2328 | 0.6643 | 0.6138 |
| 3DROT 1-1-5 | 0.1711 | 0.5714 | 0.7496 | 0.2358 | **0.6607** | **0.6178** |
| 3DROT 2-2-5 | 0.1685 | **0.5670** | **0.7548** | **0.2333** | 0.6612 | 0.612 |
| 3DROT 3-3-5 | **0.1683** | 0.5695 | 0.7536 | 0.2352 | 0.6633 | 0.6117 |
| 3DROT 2-2-10 | 0.1741 | 0.5829 | 0.7430 | 0.2375 | 0.6675 | 0.6086 |

Table 8: BTS ResNet-50 depth estimation on NYU Depth v2 and SUN RGB-D under different rotation schedules.

| | | NYU Depth v2 | | | SUN RGB-D | | |
|---|---|---|---|---|---|---|---|
| Family | Aug. | abs$\downarrow$ | rmse$\downarrow$ | a1$\uparrow$ | abs$\downarrow$ | rmse$\downarrow$ | a1$\uparrow$ |
| base | plain | 0.2105 | 0.6765 | 0.6582 | 0.2891 | 0.7587 | 0.5366 |
| proj_family | flip | 0.2027 | 0.6828 | 0.6574 | 0.2704 | **0.7408** | 0.5485 |
| proj_family | 2DRot | 0.2059 | 0.6460 | 0.6818 | 0.2835 | 0.7567 | 0.5343 |
| proj_family | 3DRot$_{2-2-5}$ | **0.1940** | **0.6370** | **0.6954** | **0.2684** | 0.7414 | **0.5494** |
| proj_family | 3DRot$_{3-3-5}$ | 0.1965 | 0.6430 | 0.6918 | 0.2733 | 0.7504 | 0.5429 |
| color_family | color | **0.1879** | **0.6102** | **0.7134** | **0.2532** | **0.6867** | **0.5932** |

Table 9: Full BTS ResNet-50 results with all augmentation variants. Models are trained on NYU Depth v2 and evaluated on NYU Depth v2 and SUN RGB-D. base uses the default BTS schedule (including 2D rotation and horizontal flips); proj_family applies projective augmentations (flip, in-plane 2D rotation baseline, or our camera-centric 3DRot$_{2-2-5}$/3DRot$_{3-3-5}$); color_family adds stronger photometric jitter.

| Exp. | Overall | | | Pedestrian | | | Cyclist | | | Car | | |
|---|---|---|---|---|---|---|---|---|---|---|---|---|
| | 3D | BEV | AOS | 3D | BEV | AOS | 3D | BEV | AOS | 3D | BEV | AOS |
| BASE | 63.85 | 70.79 | 66.60 | 55.12 | 62.90 | 45.25 | 58.13 | 61.48 | 62.01 | 78.31 | 88.00 | 92.53 |
| 3-0-0-0.3 | 62.97 | 69.63 | 66.76 | 56.25 | 63.78 | 47.74 | 56.12 | 59.38 | 60.52 | 76.55 | 85.74 | 92.02 |
| 0-3-0-0.3 | 64.87 | 71.54 | 68.96 | 57.78 | 64.32 | 49.86 | 59.32 | 63.17 | 64.50 | 77.52 | 87.13 | 92.51 |
| 0-0-3-0.3 | 58.58 | 63.84 | 53.43 | 54.98 | 59.49 | 30.92 | 45.56 | 45.98 | 38.84 | 75.21 | 86.06 | 90.52 |
| 3-3-3-0.3 | 59.38 | 67.11 | 62.62 | 47.18 | 56.20 | 41.95 | 52.90 | 57.89 | 53.71 | 78.07 | 87.25 | 92.19 |
| 1-1-0-0.3 | 62.78 | 69.68 | 67.31 | 53.30 | 60.97 | 50.35 | 56.27 | 59.73 | 59.47 | 78.77 | 88.34 | 92.10 |
| 3-3-0-0.3 | 65.16 | 71.09 | 67.18 | 59.26 | 66.40 | 46.87 | 57.94 | 59.35 | 62.44 | 78.27 | 87.50 | 92.23 |
| 5-5-0-0.3 | 63.40 | 70.75 | 67.56 | 55.54 | 65.44 | 45.50 | 53.71 | 58.09 | 60.49 | 79.20 | 88.19 | 92.03 |
| 3-3-0-0.1 | 63.50 | 70.64 | 65.53 | 54.02 | 62.39 | 40.94 | 57.57 | 60.57 | 60.56 | 78.74 | 87.92 | 92.45 |
| 3-3-0-0.5 | 63.47 | 69.71 | 67.02 | 56.43 | 63.27 | 47.90 | 55.29 | 58.00 | 60.86 | 78.70 | 87.85 | 92.29 |

Table 10: Full MVX-Net ablations on KITTI under different AugRotate3D settings (yaw–pitch–roll–probability). We report moderate $AP_{3D}$, $AP_{BEV}$, and AOS for Overall and each class.

# I  IMPACT OF APPROXIMATE INTRINSICS

This section elaborates on how approximate or heterogeneous camera intrinsics affect our augmentation and how they can be handled in practice.

**Estimating intrinsics from 2D–3D correspondences.**  In many 3D vision benchmarks, camera intrinsics are either provided or can be reasonably estimated. Whenever a task supplies 2D pixel coordinates together with corresponding 3D points, one can recover the full camera matrix from a small set of 2D–3D correspondences. The camera parameters—intrinsics, rotation, and translation—jointly have 11 degrees of freedom. Each 2D–3D correspondence contributes two independent constraints, because the projection of a 3D point carries its own unknown depth/scale. Therefore at least six non-degenerate correspondences are required ($2 \times 6 \geq 11$). If these six 3D points are non-coplanar, the constraints are independent and one can solve for the full camera matrix $P \in \mathbb{R}^{3 \times 4}$ and then decompose it as

$$P = K\,[R \mid t],$$

where $K$ is upper-triangular with positive diagonal entries and $(R, t)$ are the extrinsics.

**Effect of approximate intrinsics.**  When the estimated intrinsics $K$ are only approximate, the image warp and label update in 3DRot may become slightly mismatched. Empirically, moderate perturbations in $K$ tend to behave like a smooth re-parameterization of the camera rays: the mapping from pixels to rays in $\mathbb{R}^3$ is distorted but remains consistent across the image. Modern detectors are usually robust to such structured perturbations and can absorb them during training, so the gains from 3DRot typically degrade gracefully rather than collapsing.

**Normalizing heterogeneous intrinsics via a virtual camera.**  A related concern is that large diversity in intrinsics across a dataset might limit the applicability of camera-centric augmentations. In practice, this diversity can be absorbed by adopting a shared *virtual camera* parameterization as in Cube R-CNN (Brazil et al., 2023). Concretely, consider a real camera with intrinsics $K$ (focal length $f$) and image height $H$, and a fixed virtual camera $(K_v, f_v, H_v)$. For a 3D point with metric depth $z$ in the real camera, the corresponding *virtual depth* $z_v$ is

$$z_v = z \cdot \frac{f_v}{f} \cdot \frac{H}{H_v}.$$

This change of variables is a similarity transform in projective space. A 3D point $\mathbf{X}$ that satisfies

$$s\,\mathbf{x} = K\,[R \mid t]\,\mathbf{X}$$

in the real camera can be equivalently represented in the virtual camera as

$$s_v\,\mathbf{x} = K_v\,[R_v \mid t_v]\,\mathbf{X}_v,$$

where $\mathbf{X}_v$ encodes the same scene geometry but uses $z_v$ instead of $z$. Our augmentation only requires that 3D points and 2D pixels obey a consistent pinhole projection model. Therefore, 3DRot can be applied directly in this unified virtual-camera space: (i) normalize all annotations to the virtual camera, (ii) perform geometry-consistent 3D rotations and reflections there, and (iii) optionally map the augmented samples back to the original cameras if needed.

In summary, when exact intrinsics are unavailable, one can either estimate them from sparse 2D–3D correspondences or absorb their variability into a virtual-camera parameterization. In both cases, 3DRot remains valid as long as the chosen camera model defines a consistent mapping between image pixels and 3D rays.

## J    LLM Usage Statement

An LLM (ChatGPT, GPT-5 Thinking) was used only for language polishing, LaTeX layout sugges-tions. It did not contribute to ideas, methods, derivations, experiments, or conclusions. The authors take full responsibility for all content; the LLM is not an author.

