# OpenReview forum: "3DRot: 3D Rotation Augmentation for RGB-Based 3D Tasks"
_ICLR.cc/2026/Conference — Submitted to ICLR 2026_

### Official Review · Reviewer_kBMV · 2025-10-23

**Soundness:** 2
**Presentation:** 1
**Contribution:** 2
**Rating:** 2
**Confidence:** 5

**Summary:**

This paper proposes 3DRot, which introduces a 3D rotation augmentation for RGB-based 3D tasks by rotating/mirroring scenes around the camera's optical center while preserving projective geometry through synchronized updates of RGB images, camera intrinsics, and 3D annotations.  This method provides a data-efficient solution that significantly improved performance on monocular 3D detection, achieving gains comparable to multi-dataset training using only a single dataset.

**Strengths:**

++ The augmentation does not require depth information.

++ The 3DRot augmentation has been validated to be effective on the RGB-based 3D detection task.

**Weaknesses:**

-- While the authors acknowledge that standard flipping (reflection) augmentation violates chirality, they claim to use a chirality-preserving method. However, the methodology is only briefly mentioned in the ablation study. A detailed explanation of how chirality is preserved should be provided in the Method section to ensure clarity and readability.

-- The depths and NOCS maps in Figure 2 do not appear to be utilized in the experiments. It seems necessary to incorporate them into relevant tasks for coherence; otherwise, they may seem superfluous.

-- The claim of cross-domain generalization is not sufficiently substantiated. The experimental validation is limited to a single task and two datasets, raising concerns that the reported gains may be specific to that particular experimental setup. To robustly support this claim, the authors should demonstrate the effectiveness of their method across a wider range of tasks and datasets. Furthermore, the potential need to re-tune parameters of the augmentation strategy for different domains remains an open question.

-- Overly simple derivations and basic formulas can be omitted from the main text to free up space for experimental details.

-- A limitation of the study is the absence of a comparison with other augmentation techniques like 3D copy-paste, both in isolation and combined. This omission makes it difficult to discern the specific advantages of the proposed method.


-- A discussion of the limitations and potential failure cases of the proposed augmentation method should be included.

Minor:

-- The dimensionality is usually denoted using \mathbb{R}, but this paper uses several different (and inconsistent) notations.

**Questions:**

-- The method relies on camera intrinsic parameters. How would its performance be affected by approximated or inaccurate camera intrinsics?

---

> ### Author Response · Authors · 2025-11-24
> **Response to concerns on chirality, depth/NOCS usage, and evaluation scope**
>
> We thank you for your detailed and critical assessment. We appreciate your concerns about validation scope, chirality, and presentation, and have substantially revised the manuscript accordingly.
>
> ---
>
> ### 1. Short recap of the revised paper
>
> The revised paper introduces **3DRot**, a camera-centric rotation/flip augmentation that
> (1) rotates or mirrors RGB images around the **optical center**,
> (2) synchronously updates intrinsics, 3D poses, depth maps, and (when available) LiDAR points,
> (3) preserves 2D–3D projective geometry **without requiring scene depth**.
>
> We now validate 3DRot in **three settings**:
>
> - **Monocular 3D cuboid detection** on SUN10 / IN10 (frozen DINO-X + Cube R-CNN； IoU$_{3D}$↑ improves from 43.21→44.51 and ROT error↓ from 22.91°→20.93°).
> - **Monocular depth estimation** with BTS ResNet-50 on NYU Depth v2 → SUN RGB-D (NYU abs-rel↓ 0.1783→0.1685; SUN RGB-D abs-rel↓ 0.2502→0.2333).
> - **LiDAR+RGB 3D detection** with MVX-Net on KITTI(moderate Overall AP$_{3D}$↑ 63.85→65.16).
>
>
> ---
>
> ### 2. Chirality-preserving flips
>
> We agree that chirality preservation must be clearly explained in the Method, not only via ablations.
>
> - We add a dedicated **“Flipping Transformation”** subsection: we reflect around the optical center, then re-orthogonalize the camera basis and enforce a right-handed frame by negating the third basis vector, so the final rotation remains in \(SO(3)\) and avoids chirality-flipping ambiguities.
> - Detailed derivations are moved to an appendix, keeping the main text concise.
>
> ---
>
> ### 3. Depth / NOCS maps and their use
>
> We agree that the role of depth and NOCS should be tied to concrete experiments.
>
> - We now **explicitly use the same camera-centric mapping in a depth-estimation task** by inserting 3DRot into **BTS ResNet-50** trained on **NYU Depth v2**.
>   - On NYU: abs-rel↓ 0.1783→0.1685, δ<1.25↑ 0.7472→0.7548.
>   - On SUN RGB-D: abs-rel 0.2502→0.2333, rmse also decreases.
> - The figure caption and nearby text are updated to state that the depth / NOCS visualizations depict exactly this camera-centric mapping for dense maps, not only for cuboids.
>
> ---
>
> ### 4. Cross-domain generalization, scope, and tuning
>
> We agree that the original submission did not fully support cross-domain claims.
>
> - We **broaden the evaluation** to three tasks and multiple datasets:
>   1. **Monocular 3D detection:** SUN10 and cross-domain IN10 (SUN RGB-D + ARKitScenes + Hypersim). 3DRot improves all seven pose metrics on SUN10, with smaller but consistent gains on IN10.
>   2. **Monocular depth estimation:** NYU Depth v2 → SUN RGB-D, as above, improving both in-domain and cross-dataset metrics.
>   3. **LiDAR+RGB 3D detection (KITTI):** a small yaw+pitch setting of 3DRot improves moderate Overall AP\(_{3D}\) from 63.85→65.16 without harming Car AP.
>
> ---
>
> ### 5. Overly simple derivations vs. experimental details
>
> We followed your suggestion to prioritize experimental clarity.
>
> - We **shorten the Method section** by moving basic derivations (world-to-camera algebra, padding details, step-by-step affine proofs) to the appendix.
> - The main text now keeps only the geometric intuition, the final update rules, and a brief explanation of chirality-preserving flips, freeing space for the new experiments and discussion.
>
> ---
>
> ### 6. Comparison to other augmentation techniques
>
> Because of the per-comment character limit, we only give a **high-level response** here and will provide the **full augmentation comparison tables** in a short separate comment titled *“Augmentation comparison details for this review”*.
>
> At a high level:
>
> - Across SUN10/IN10 detection, NYU/SUN depth, and KITTI MVX-Net, we explicitly compare 3DRot with standard horizontal flips, in-plane 2D rotation, and stronger photometric jitter.
> - Within the **projective-geometry family** (flip / 2D rotation / 3DRot), 3DRot consistently gives the best trade-off on the main metrics in each task, while color jitter is complementary rather than a replacement.
>
> ---
>
> ### 7. Limitations and potential failure cases
>
> We now add an explicit **“Conclusion & Discussion”** paragraph on limitations.
>
> ---
>
> ### 8. Minor – Dimensionality notation
>
> We have **standardized all dimensionality notation** to the usual \(\mathbb{R}^{\cdot}\) form throughout the paper and removed inconsistent variants.
>
> ---
>
> ### 9. Question – Effect of approximate or inaccurate intrinsics
>
> You are right that 3DRot requires camera intrinsics to guarantee 2D–3D consistency.
>
> When intrinsics are not given, they can be estimated from a small number of 2D–3D correspondences. When intrinsics vary strongly across the dataset, we follow Cube R-CNN and map all views to a shared **“virtual camera”** so that 3DRot can be applied in this normalized space. We have clarified these applicability conditions in the revised text.
>
> ---
>
> We thank you again for the careful and critical review.

---

> ### Author Response · Authors · 2025-11-24
> **Augmentation comparison details for this review**
>
> We now include cross-task comparisons of different augmentation families (for monocular 3D cuboid detection, monocular depth estimation, and LiDAR+RGB 3D detection):
>
> **Table 1: Validation results with different augmentations (cuboid detection).**
>
> | Method              | IoU$_{3D}\\uparrow$ | Trans (cm)$\downarrow$ | Rot (deg)$\downarrow$ | Size (cm)$\downarrow$ | 5GRAV$\uparrow$ | 10HEAD$\uparrow$ |
> | ------------------- | -------------------- | ---------------------- | --------------------- | --------------------- | --------------- | ---------------- |
> | BASE                | 42.43                | 11.87                  | 23.99                 | 13.09                 | 77.03           | 45.39            |
> | BASE + Color jitter | 41.51                | 12.24                  | 22.48                 | 13.07                 | **78.84**       | 48.16            |
> | BASE + 3DRot        | **44.41**            | **11.13**              | **21.91**             | **12.50**             | 77.26           | **49.06**        |
>
> *Trans and Size are reported in centimeters; Rot is the mean absolute rotation error in degrees. 5GRAV denotes accuracy within $5^\circ$ of the ground-truth gravity axis, and 10HEAD denotes accuracy within $10^\circ$ of the ground-truth heading. All configurations are trained without any flipping augmentation.*
>
> ---
>
> **Table 2: Full BTS ResNet-50 results with all augmentation variants (monocular depth).**
>
> | Family         | Aug.                | NYU abs$\downarrow$ | NYU rmse$\downarrow$ | NYU a1$\uparrow$ | SUN abs$\downarrow$ | SUN rmse$\downarrow$ | SUN a1$\uparrow$ |
> | -------------- | ------------------- | ------------------- | -------------------- | ---------------- | ------------------- | -------------------- | ---------------- |
> | `base`         | plain               | 0.2105              | 0.6765               | 0.6582           | 0.2891              | 0.7587               | 0.5366           |
> | `proj_family`  | flip                | 0.2027              | 0.6828               | 0.6574           | 0.2704              | **0.7408**           | 0.5485           |
> | `proj_family`  | 2DRot               | 0.2059              | 0.6460               | 0.6818           | 0.2835              | 0.7567               | 0.5343           |
> | `proj_family`  | 3DRot$_{2{-}2{-}5}$ | **0.1940**          | **0.6370**           | **0.6954**       | **0.2684**          | 0.7414               | **0.5494**       |
> | `proj_family`  | 3DRot$_{3{-}3{-}5}$ | 0.1965              | 0.6430               | 0.6918           | 0.2733              | 0.7504               | 0.5429           |
> | `color_family` | color               | **0.1879**          | **0.6102**           | **0.7134**       | **0.2532**          | **0.6867**           | **0.5932**       |
>
> *Models are trained on NYU Depth v2 and evaluated on NYU Depth v2 and SUN RGB-D. `base` uses the default BTS schedule (including 2D rotation and horizontal flips. 2DRot, treating depth as a fourth image channel but without enforcing a true roll around the principal point); `proj_family` applies projective augmentations (flip, in-plane 2D rotation baseline, or our camera-centric 3DRot$_{2{-}2{-}5}$/3DRot$_{3{-}3{-}5}$); `color_family` adds stronger photometric jitter.*
>
> ---
>
> **Table 3: Effect of $(G,F,R)$ configurations in MVX-Net ablations on KITTI (LiDAR+RGB 3D detection).**
>
> | $(G,F,R)$ | 3D box    | BEV       | AOS       |
> | --------- | --------- | --------- | --------- |
> | (0,0,0)   | 44.78     | 53.91     | 50.08     |
> | (1,0,0)   | 61.86     | 68.96     | 66.69     |
> | (0,1,0)   | 51.00     | 59.06     | 54.61     |
> | (0,0,1)   | 50.80     | 59.65     | 54.58     |
> | (1,1,0)   | 63.62     | 70.53     | 66.63     |
> | (0,1,1)   | 56.45     | 63.47     | 62.59     |
> | (1,0,1)   | 61.18     | 68.85     | 64.55     |
> | (1,1,1)   | **65.16** | **71.09** | **67.18** |
>
> *$(G,F,R)$ indicates whether GlobalRotScaleTrans (G), RandomFlip3D (F), and AugRotate3D (R) are enabled $(0/1)$. This table summarizes the interaction between these 3D augmentations for MVX-Net on KITTI (moderate AP metrics). GlobalRotScaleTrans is a pure point-cloud augmentation and does not modify the 2D images.*

---

> > ### Comment · Reviewer_kBMV · 2025-11-28
> >
> > Thank you for the response. Could you please update the manuscript with the revisions highlighted (e.g., using colored text)? This will help the reviewers efficiently identify and evaluate the modifications.

---

> > > ### Author Response · Authors · 2025-11-28
> > > **Clarification on highlighted revisions in the revised manuscript**
> > >
> > > Thank you very much for the helpful suggestion.
> > >
> > > We have updated the manuscript so that all new or modified content is highlighted in blue. In the appendix, section titles shown in blue indicate sections that are new or have been substantially revised (either moved from the main paper or added to report new experimental results).
> > >
> > > Based on your detailed earlier comments, we made many changes to improve the clarity and scope of the paper, and we are very grateful for your constructive feedback.

---

> > > > ### Comment · Reviewer_kBMV · 2025-11-28
> > > >
> > > > It seems the new experiments only use depth, and the NOCS information is still not used by any primary experiments. Therefore, it is inappropriate to depict NOCS in the main paper. Maybe it is more suitable to just put the NOCS images in the appendix.

---

> ### Author Response · Authors · 2025-11-28
> **Update on depth/NOCS figures**
>
> Thank you very much for the helpful suggestions.
>
> We have updated the manuscript accordingly: (i) the main paper now only shows the concept figure and the depth visualization, (ii) the NOCS visualizations have been moved to the appendix (Section A) together with a brief explanation, and (iii) all new or modified content is highlighted in blue for easier inspection.
>
> We agree that this organization keeps the main text focused on the evaluated tasks (detection, depth, and LiDAR+RGB), while still documenting in the appendix that 3DRot is compatible with NOCS-style outputs. Thank you again for your feedback.

---

### Official Review · Reviewer_JwxW · 2025-10-29

**Soundness:** 3
**Presentation:** 2
**Contribution:** 2
**Rating:** 4
**Confidence:** 4

**Summary:**

The paper proposes a geometric image augmentation for RGB-based 3D vision tasks (e.g., 3D detection, 3D keypoint estimation). The augmentation rotates the camera around its optical center in a way that preserves projective geometry and therefore the 2D–3D correspondences between images and 3D labels. The authors give a mathematical derivation of the augmentation and present experiments showing improved performance on indoor 3D detection benchmarks when used with a Cube R-CNN detector and a DINO-X backbone.

**Strengths:**

-	Rigorous derivation: The mathematical formulation that guarantees preservation of 2D–3D correspondences is clean and convincing.
-	Empirical ablation: The paper includes ablations about flipping/rotation configurations and shows that the proposed rotation improves accuracy compared to no-rotation.
-	Conceptual simplicity: The idea is intuitive and could be widely useful if shown to be robust across models and datasets.

**Weaknesses:**

Limited Evaluation:

- The authors did not compare their method with alternative augmentation strategies. Only the relative gain against no augmentation is reported. Other augmentations could include non-geometric augmentation (jittering) or geometric ones as discussed in related work.
- The biggest gain in performance in Table 3a) is achieved by leveraging DINO-X as the backbone of Cube R-CNN which is not the contribution of this paper.
- The augmentation strategy is only evaluated on one model (DINO-X + Cube R-CNN).

Limited Contribution:

-	Based on the conducted evaluation, the contribution of this work seems to be limited to the community.

**Questions:**

-	Is there a reason that 3DRot is only evaluated on indoor scenes? Does the technique also apply for outdoor scenes (e.g. KITTI dataset) for example?
-	What are the boundaries for the rotation angles? As their absolute value increases, more and more pixels are padded, hindering model’s training.

---

> ### Author Response · Authors · 2025-11-24
> **Response on Evaluation Scope and New Experiments**
>
> We appreciate your constructive and detailed feedback. Your comments on evaluation breadth, comparisons to other augmentations, and indoor-only settings were very helpful for improving the paper. Below we briefly summarize the main changes in the revised version and then respond to your specific concerns.
>
> ---
>
> ### 1. What changed since the submission (high level)
>
> **(1) Broader evaluation beyond one model and task.**
> In addition to the original indoor monocular 3D cuboid detection (DINO-X + Cube R-CNN on SUN10/IN10), we now evaluate 3DRot in two further settings:
>
> - **Monocular depth estimation (BTS, NYU Depth v2 → SUN RGB-D; abs-rel↓ 0.1783→0.1685 on NYU and 0.2502→0.2333 on SUN RGB-D).**
>   We insert 3DRot into BTS ResNet-50, using the standard NYU training protocol. With 3DRot on top of the default BTS augmentations (which already include 2D rotation and flips), both NYU and cross-dataset SUN RGB-D metrics improve (see Tables 4–5 in the revised paper).
>
> - **LiDAR+RGB 3D detection (MVX-Net, KITTI; moderate Overall AP $_{3D}$↑ 63.85→65.16).**
>   We plug 3DRot into MVX-Net in MMDetection3D on KITTI (Car / Pedestrian / Cyclist). A small camera-centric yaw+pitch schedule improves moderate Overall 3D AP over the baseline while keeping Car AP essentially unchanged (Table 6).
>
> Together with the original SUN10/IN10 experiments, 3DRot is now tested on **three distinct pipelines** (monocular cuboid detection, monocular depth, and LiDAR+RGB detection) with different architectures and supervision signals.
>
> **(2) Clarified implementation details.**
> We clarify that we use **small pitch/yaw (≈3–5°) and moderate roll** on SUN10/IN10 and NYU/SUN, and small yaw/pitch without roll on KITTI. We also describe a simple padding + principal-point realignment scheme that keeps all valid pixels while maintaining a consistent intrinsic matrix (Section 3.5 and Implementation Details).
>
> ---
>
> ### 2. Response to “Limited evaluation” and “Limited contribution”
>
> > “The authors did not compare their method with alternative augmentation strategies… The augmentation strategy is only evaluated on one model (DINO-X + Cube R-CNN)… the contribution seems limited to the community.”
>
> Because of the per-comment character limit, we only give a **high-level response** here and will provide the **full augmentation comparison tables** (baseline vs flip vs 2D rotation vs 3DRot vs stronger color jitter) in a short separate comment titled *“Augmentation comparison details for this review”*.
>
> At a high level:
>
> - Across SUN10/IN10 detection, NYU/SUN depth, and KITTI MVX-Net, we explicitly compare 3DRot with standard horizontal flips, in-plane 2D rotation, and stronger photometric jitter.
> - Within the **projective-geometry family** (flip / 2D rotation / 3DRot), 3DRot consistently gives the best trade-off on the main metrics in each task, while color jitter is complementary rather than a replacement.
> - In all three pipelines we keep the backbone and training recipe fixed and only add 3DRot on top, so the reported gains reflect the contribution of the augmentation itself rather than architectural changes.
>
> ---
>
> ### 3. Indoor-only evaluation vs outdoor scenes
>
> > “Is there a reason that 3DRot is only evaluated on indoor scenes? Does the technique also apply for outdoor scenes (e.g., KITTI)?”
>
> Our initial focus on indoor datasets was due to their diverse roll/pitch/yaw, which stresses rotation robustness. In response to your question, we have added **outdoor experiments on KITTI** with MVX-Net:
>
> - Small yaw+pitch rotations about the optical center improve moderate Overall 3D AP on KITTI.
> - Larger roll angles hurt Cyclist/Overall AP, which we attribute to the acquisition setup (cameras roughly level w.r.t. the ground), making large roll unrealistic.
>
> We now explicitly discuss this dataset-dependent behavior in the conclusion: 3DRot is not restricted to indoor scenes, but the **useful angle ranges depend on the camera pose distribution** of each dataset.
>
> ---
>
> ### 4. Rotation angle bounds and padding/cropping
>
> > “What are the boundaries for the rotation angles? As their absolute value increases, more and more pixels are padded, hindering model’s training.”
>
> In practice we use **small-angle rotations**:
>
> - Indoor (SUN10/IN10, NYU/SUN): pitch/yaw up to about 3–5°, roll up to about 0–10°.
> - Outdoor (KITTI): small yaw/pitch without roll in the best configuration.
>
> To avoid discarding valid pixels, we do not simply rotate and crop back to the original frame. Instead, we warp the image corners, take the minimal enclosing rectangle, render the rotated image on this canvas, and update the principal point accordingly. This keeps all valid pixels and ensures that the updated intrinsics remain consistent with the rendered image.
>
> We hope these additional experiments and clarifications address your concerns about evaluation scope, applicability to outdoor scenes, and the practical behavior of 3DRot.

---

> ### Author Response · Authors · 2025-11-24
> **Augmentation comparison details for this review**
>
> We now include cross-task comparisons of different augmentation families (for monocular 3D cuboid detection, monocular depth estimation, and LiDAR+RGB 3D detection):
>
> **Table 1: Validation results with different augmentations (cuboid detection).**
>
> | Method              | IoU$_{3D}\\uparrow$ | Trans (cm)$\downarrow$ | Rot (deg)$\downarrow$ | Size (cm)$\downarrow$ | 5GRAV$\uparrow$ | 10HEAD$\uparrow$ |
> | ------------------- | -------------------- | ---------------------- | --------------------- | --------------------- | --------------- | ---------------- |
> | BASE                | 42.43                | 11.87                  | 23.99                 | 13.09                 | 77.03           | 45.39            |
> | BASE + Color jitter | 41.51                | 12.24                  | 22.48                 | 13.07                 | **78.84**       | 48.16            |
> | BASE + 3DRot        | **44.41**            | **11.13**              | **21.91**             | **12.50**             | 77.26           | **49.06**        |
>
> *Trans and Size are reported in centimeters; Rot is the mean absolute rotation error in degrees. 5GRAV denotes accuracy within $5^\circ$ of the ground-truth gravity axis, and 10HEAD denotes accuracy within $10^\circ$ of the ground-truth heading. All configurations are trained without any flipping augmentation.*
>
> ---
>
> **Table 2: Full BTS ResNet-50 results with all augmentation variants (monocular depth).**
>
> | Family         | Aug.                | NYU abs$\downarrow$ | NYU rmse$\downarrow$ | NYU a1$\uparrow$ | SUN abs$\downarrow$ | SUN rmse$\downarrow$ | SUN a1$\uparrow$ |
> | -------------- | ------------------- | ------------------- | -------------------- | ---------------- | ------------------- | -------------------- | ---------------- |
> | `base`         | plain               | 0.2105              | 0.6765               | 0.6582           | 0.2891              | 0.7587               | 0.5366           |
> | `proj_family`  | flip                | 0.2027              | 0.6828               | 0.6574           | 0.2704              | **0.7408**           | 0.5485           |
> | `proj_family`  | 2DRot               | 0.2059              | 0.6460               | 0.6818           | 0.2835              | 0.7567               | 0.5343           |
> | `proj_family`  | 3DRot$_{2{-}2{-}5}$ | **0.1940**          | **0.6370**           | **0.6954**       | **0.2684**          | 0.7414               | **0.5494**       |
> | `proj_family`  | 3DRot$_{3{-}3{-}5}$ | 0.1965              | 0.6430               | 0.6918           | 0.2733              | 0.7504               | 0.5429           |
> | `color_family` | color               | **0.1879**          | **0.6102**           | **0.7134**       | **0.2532**          | **0.6867**           | **0.5932**       |
>
> *Models are trained on NYU Depth v2 and evaluated on NYU Depth v2 and SUN RGB-D. `base` uses the default BTS schedule (including 2D rotation and horizontal flips. 2DRot, treating depth as a fourth image channel but without enforcing a true roll around the principal point); `proj_family` applies projective augmentations (flip, in-plane 2D rotation baseline, or our camera-centric 3DRot$_{2{-}2{-}5}$/3DRot$_{3{-}3{-}5}$); `color_family` adds stronger photometric jitter.*
>
> ---
>
> **Table 3: Effect of $(G,F,R)$ configurations in MVX-Net ablations on KITTI (LiDAR+RGB 3D detection).**
>
> | $(G,F,R)$ | 3D box    | BEV       | AOS       |
> | --------- | --------- | --------- | --------- |
> | (0,0,0)   | 44.78     | 53.91     | 50.08     |
> | (1,0,0)   | 61.86     | 68.96     | 66.69     |
> | (0,1,0)   | 51.00     | 59.06     | 54.61     |
> | (0,0,1)   | 50.80     | 59.65     | 54.58     |
> | (1,1,0)   | 63.62     | 70.53     | 66.63     |
> | (0,1,1)   | 56.45     | 63.47     | 62.59     |
> | (1,0,1)   | 61.18     | 68.85     | 64.55     |
> | (1,1,1)   | **65.16** | **71.09** | **67.18** |
>
> *$(G,F,R)$ indicates whether GlobalRotScaleTrans (G), RandomFlip3D (F), and AugRotate3D (R) are enabled $(0/1)$. This table summarizes the interaction between these 3D augmentations for MVX-Net on KITTI (moderate AP metrics). GlobalRotScaleTrans is a pure point-cloud augmentation and does not modify the 2D images.*

---

### Official Review · Reviewer_c3Pg · 2025-11-01

**Soundness:** 4
**Presentation:** 3
**Contribution:** 3
**Rating:** 6
**Confidence:** 3

**Summary:**

This paper presents a data augmentation method that work on the rotation of the viewing direction of the camera. The method rotates and mirrors images about the camera's optical center. The method does not need depth information. The author introduces the math foundations for the transformations of pixel locations based on which the remap is defined. The data augmentation method is evaluated by the performance gain of the 3D detection task.

**Strengths:**

- The idea of augmenting the data using camera view direction change is a nice and interesting idea.
- The authors presented detailed math derivation to backup the method.
- The method does not need depth information.
- The performance gain is notable on 3D detection task on various datasets.

**Weaknesses:**

- I would suggest the authors to take a closer look at related works. It would be quite surprising if the same or similar idea has never been used by others. It is likely that people already used this augmentation method for quite a long time yet did not publish it because it seems obvious and trivial. I would suggest the authors put more effort in investigating the prior works.
- Though depth information is not required, this method still needs camera intrinsics. Such information may not be available for random image data. So this may limited the scope of this method being useful.
- I am not sure if 3D detection is a good task for evaluating the data augmentation method, since the performance gain seems to be modest. Would other tasks be better where the performance gain is more significant?

**Questions:**

- Roll rotation seems to be OK which is what vanilla random rotation augmentation already did. But yaw and pitch may have some problems. The principal point is supposed to be at the center of any image. But after changing the view direction, the principal point will not be at the center. How did the authors deal with such situation? Maybe this has been mentioned in the paper but I may have missed it.

---

> ### Author Response · Authors · 2025-11-24
> **Clarifications on 3DRot novelty, experiments, and implementation**
>
> Thank you very much for the positive and constructive review, and for highlighting both the interesting nature of our camera-centric augmentation and the concerns about related work and evaluation scope.
>
> ---
>
> ### Recap and new experiments
>
> Our paper introduces **3DRot**, a plug-and-play *camera-centric* augmentation that rotates and mirrors RGB images about the optical center while synchronously updating RGB, camera intrinsics, and 3D labels so that 2D–3D projection remains exactly consistent, without requiring scene depth. In the original submission we mainly evaluated monocular 3D cuboid detection on SUN RGB-D / IN10.
>
> Motivated directly by your comments, we have substantially broadened the evaluation and clarified the implementation:
>
> * **New task 1 – Monocular depth estimation (BTS, NYU Depth v2 → SUN RGB-D).**
>   We insert 3DRot into BTS ResNet-50. On NYU Depth v2, abs-rel↓ improves from **0.1783 → 0.1685** and δ<1.25↑ from **0.7472 → 0.7548**. On cross-dataset SUN RGB-D, abs-rel↓ improves from **0.2502 → 0.2333** and rmse↓ from **0.6738 → 0.6612** (Table 4 and Table 5).
> * **New task 2 – LiDAR+RGB 3D detection (MVX-Net, KITTI).**
>   We plug 3DRot into MVX-Net in MMDetection3D. A small camera-centric yaw+pitch schedule (R₃₋₃₋₀) improves moderate Overall AP₃D↑ from **63.85 → 65.16** while keeping Car AP essentially unchanged, and remains compatible with GlobalRotScaleTrans and RandomFlip3D (Table 6).
>
> ---
>
> ### (1) Novelty and related work
>
> We agree that the idea of camera-space rotation is natural and have expanded the related-work section to make the differences precise; due to the character limit, we only summarize the key points here and provide additional details in a separate comment.
>
> Most prior RGB-based augmentations either
> (i) rely on **depth** and explicitly reproject RGB-D into new views,
> (ii) rely on **ground-plane homographies** (thus assume coplanarity), or
> (iii) focus on in-plane that are implemented as 2D image warps with a yaw correction on the 3D pose; these methods report visually plausible 2D–3D consistency but do not provide a general geometric analysis.
>
> In contrast, 3DRot:
>
> * uses only a calibrated pinhole camera, **without any depth maps or ground-plane assumption**;
> * derives a **closed-form camera-space homography** that simultaneously updates RGB, intrinsics, and full 3×3 rotations for arbitrary 3D scenes;
> * naturally extends to **multi-modal signals (depth maps, LiDAR points)** via the same camera-centric transform.
>
> To our knowledge, existing work does not provide this general optical-center formulation together with a systematic evaluation across **monocular detection, monocular depth, and LiDAR+RGB detection**.
>
> ---
>
> ### (2) On task choice and magnitude of gains
>
> On **SUN10** monocular 3D detection, inserting 3DRot into a frozen DINO-X + Cube R-CNN pipeline improves IoU₃D↑ from **43.21 → 44.51**, reduces rotation error from **22.91° → 20.93°**, and increases mAP₀.₅ from **35.70 → 38.11** (Tables 1–3). For a plug-and-play augmentation on top of a strong off-the-shelf detector, we believe a +1.3 IoU₃D and +2.4 mAP₀.₅ gain is meaningful and comparable to gains reported for stronger training domains.
>
> Your suggestion that 3D detection may not be the best task for evaluation was very helpful. In response, we added:
>
> * **Monocular depth estimation**, where 3DRot improves both in-domain and cross-dataset metrics on top of a baseline that already uses 2D rotations and flips (Tables 4–5); and
> * **LiDAR+RGB detection (MVX-Net)**, where 3DRot provides extra pose diversity and remains stable when combined with standard 3D scene-level augmentations (Table 6).
>
> ---
>
> ### (3) Dependence on camera intrinsics
>
> You are right that 3DRot requires camera intrinsics to guarantee 2D–3D consistency.
>
> When intrinsics are not given, they can be estimated from a small number of 2D–3D correspondences. When intrinsics vary strongly across the dataset, we follow Cube R-CNN and map all views to a shared **“virtual camera”** so that 3DRot can be applied in this normalized space. We have clarified these applicability conditions in the revised text.
>
> ---
>
> ### (4) Principal point under yaw/pitch rotations
>
> We appreciate this insightful question. Indeed, the principal point does change under yaw/pitch rotations. We briefly describe this in Section 3.5, with full algorithmic details given in Appendix D.
>
> 1. Rotate the view around the optical center and project the image corners into the rotated frame.
> 2. Compute the minimal axis-aligned bounding box that contains all rotated corners.
> 3. Render the rotated image on this canvas and set the new principal point to the canvas center.
>
> ---
>
> We again appreciate your thoughtful comments. They led us to broaden the experimental scope, clarify the geometry and assumptions, and strengthen the positioning of 3DRot as a general camera-space augmentation primitive for RGB-based and multi-modal 3D tasks.

---

> ### Author Response · Authors · 2025-11-24
> **About novelty and related work coverage:**
>
> We also found it surprising that this seemingly natural camera-space augmentation has not been systematically studied. To our knowledge, prior work on RGB-based 3D tasks does not provide a general, geometry-consistent formulation of such rotations.
>
> A work on data augmentation in the 3D field: [Exploring Geometric Consistency for Monocular 3D Detection](https://arxiv.org/abs/2104.05858)(CVPR’22) explicitly notes that, for a long time, horizontal flip and color distortion were essentially the only augmentations used in this setting (p. 3, Sec. 2.2).
>
> We did find an appendix note A.4 in [CARLA Drone: Monocular 3D Object Detection from a Different Perspective](https://arxiv.org/pdf/2408.11958)(GCPR'24) observing that small roll rotations can “look” compatible with detections, but without math or controlled experiments.
>
> Recent systems such as [Cube R-CNN](https://arxiv.org/pdf/2207.10660) (CVPR’23) , for example, uses horizontal flipping as described in the experimental implementation details, but does not include optical-center rotations. And [NOCSformer](https://arxiv.org/abs/2407.08711) (ECCV’24) only mentions random image resizing in Appendix C.1 (‘Data augmentation’).
>
> Moreover, this rotation data augmentation is not included in the ‘Customize Data Pipelines’ tutorial of the widely used [MMDetection3D](https://mmdetection3d.readthedocs.io/en/v0.17.1/tutorials/data_pipeline.html) framework.
>
> Our main technical observation is that rotating around the optical axis avoids the coplanarity constraint inherent in homography-based formulations. This property, to our knowledge, has not been explicitly exploited in prior RGB-based 3D augmentation work.
>
> [Monocular 3D object detection for an indoor robot environment](https://ieeexplore.ieee.org/stamp/stamp.jsp?tp=&arnumber=9223480)(RO-MAN'20) proposes a rotational data augmentation method that ***\*requires scene depth information\****. They utilize the scene's depth information and then reproject it from a new perspective to obtain a 2D image of that new viewpoint.
>
> Compared to augmentation strategies that rely on additional generative or reconstruction models (e.g., 3D copy-paste), our camera-space formulation is lightweight and fully interpretable: it explicitly preserves 3D–2D projection consistency. We believe this simple geometric perspective can be useful to the community.
>
> Rotation augmentation is indeed common in pure point-cloud methods, where 2D consistency is not a concern, so there is no need to consider 3D-2D projection consistency. However, when performing cross-modal point cloud tasks, maintaining 3D-2D projection consistency becomes a problem, as mentioned in section related work  augmentations for detection of [Exploring Data Augmentation for Multi-Modality 3D Object Detection](https://arxiv.org/pdf/2012.12741)(CVPR'21). Both Exploring Data Augmentation for Multi-Modality 3D Object Detection and [PointAugmenting: Cross-Modal Augmentation for 3D Object Detection](https://arxiv.org/pdf/2012.12741)(CVPR'21) only mention random cropping, random flipping, multi-scale training, and copy-paste for multi-modality methods as common augmentation techniques.

---

### Official Review · Reviewer_8Fwf · 2025-11-04

**Soundness:** 3
**Presentation:** 3
**Contribution:** 2
**Rating:** 4
**Confidence:** 3

**Summary:**

This paper proposes a 3D rotation augmentation for monocular 3D object detection. The main proposed design is to apply 3D rotations and reflections on the camera's optical center, and update the camera intrinsics, 3D annotations, and RGB images accordingly. Experimental results are conducted on the 3D object detection task to validate the effectiveness of the proposed method.

**Strengths:**

- The proposed 3D rotation augmentation is reasonable and supported by theoretical derivations and proof.
- Experimental results on the 3D detection task show the effectiveness of the proposed augmentation.
- The paper is well written and easy to follow.

**Weaknesses:**

- The experimental evaluation of the proposed augmentation is not thorough:
  - The paper claims the augmentation for RGB-based 3D tasks, but only the 3D object detection task is evaluated. Evaluation on more tasks, e.g., monocular depth estimation, would be beneficial.
  - The 3D object detection is only conducted on 10 categories and only on indoor SUN RGB-D. Evaluation on more diverse categories and larger datasets that include both indoor and outdoor scenes will improve the validation of the generalizability of the proposed method.
  - Only one algorithm framework is utilized for the benchmark. Can this augmentation benefit more 3D detection frameworks?
- Considering that more and more large-scale and diverse datasets are proposed, e.g., ScanNet, Matterport3D, and ARKitScenes, it's not clear whether the proposed augmentation still provides a large performance gain in these larger-scale real-world datasets.

**Questions:**

See the weakness section.

---

> ### Author Response · Authors · 2025-11-24
> **Broader evaluation with depth and LiDAR experiments**
>
> Thank you for the thoughtful and constructive feedback.
>
> Our paper introduces **3DRot**, a plug-and-play camera-centric rotation and reflection augmentation that operates around the optical center and synchronously updates RGB, intrinsics, and 3D labels without using scene depth. Your comments about evaluation scope were very helpful, and in the revision we have **substantially expanded the experiments**:
>
> - We add a **monocular depth estimation benchmark** (BTS ResNet-50 on NYU Depth v2 → SUN RGB-D. **abs-rel↓ 0.178→0.168** on NYU and **0.25→0.233** on SUN RGB-D).
> - We add a **LiDAR+RGB 3D detection benchmark** (MVX-Net on KITTI; **moderate Overall AP $_{3D}$↑  63.85→65.16**) to cover multi-modal and outdoor settings.
> - We extend **ablations and dataset coverage** for monocular detection (SUN10/IN10) and clarify the setup.
>
> Below we address each of your concerns.
>
> ---
>
> ### R1. Only one RGB-based 3D task
>
> > “The paper claims the augmentation for RGB-based 3D tasks, but only the 3D object detection task is evaluated. Evaluation on more tasks, e.g., monocular depth estimation, would be beneficial.”
>
> We fully agree and have added **two additional tasks** beyond monocular 3D detection:
>
> - **Monocular depth (NYU Depth v2 → SUN RGB-D).**
>   We insert 3DRot into a BTS ResNet-50 model trained on NYU Depth v2. On top of the default BTS augmentation schedule (which already includes 2D rotation and horizontal flips), 3DRot:
>   - reduces NYU abs-rel↓ from **0.1783 → 0.1685** and improves δ < 1.25↑ from **0.7472 → 0.7548**, and
>   - reduces SUN RGB-D abs-rel↓ from **0.2502 → 0.2333** and improves δ < 1.25↑ from **0.6049 → 0.612**.
>     Within the “projective” family, 3DRot also outperforms pure 2D rotation and horizontal flips (Tables 5, 7–8 in the revision).
>
> - **LiDAR+RGB 3D detection (KITTI, MVX-Net).**
>   We plug 3DRot into MVX-Net in MMDetection3D. A small yaw+pitch configuration improves moderate Overall AP₃D↑ from **63.85 → 65.16** without degrading Car AP, and remains compatible with standard 3D augmentations such as GlobalRotScaleTrans and RandomFlip3D (Tables 6 and 10).
>
> ---
>
> ### R2. Only 10 categories and only indoor SUN RGB-D
>
> > “The 3D object detection is only conducted on 10 categories and only on indoor SUN RGB-D… Evaluation on more diverse categories and larger datasets… will improve the validation of the generalizability.”
>
> We have expanded both **indoor** and **outdoor** coverage:
>
> - **Indoor diversity via IN10.**
>   We already evaluate on **IN10**, which merges SUN RGB-D, ARKitScenes, and Hypersim into a cross-domain 10-class split. 3DRot yields consistent gains (e.g., 3D IoU↑ **30.39 → 30.59**, translational error **22.97 → 22.75 cm**).
>
> - **Outdoor diversity via KITTI.**
>   In the revision, we add **KITTI 3D detection** with MVX-Net (LiDAR+RGB). This introduces outdoor driving scenes and different camera rigs; 3DRot still improves moderate AP₃D↑ (Overall **63.85 → 65.16**).
>
> Since 3DRot operates purely in camera space (center, depth, rotation, size), its behavior is largely orthogonal to the specific class vocabulary.
>
> ---
>
> ### R3. Only one detection framework
>
> > “Only one algorithm framework is utilized for the benchmark. Can this augmentation benefit more 3D detection frameworks?”
>
> In the revision we validate 3DRot across **three distinct architectures and tasks**:
>
> - **Frozen DINO-X + Cube R-CNN** for monocular 3D cuboid detection (SUN10/IN10),
> - **BTS ResNet-50** for monocular depth estimation (NYU Depth v2 → SUN RGB-D),
> - **DynamicMVXFasterRCNN (MVX-Net)** for LiDAR+RGB 3D detection (KITTI).
>
> In all three cases, inserting 3DRot as a pre-processing operator yields consistent improvements on the main 3D metrics, without changing the network architecture or loss. This supports our claim that 3DRot behaves like a **framework-agnostic augmentation primitive** that can be dropped into existing pipelines.
>
> ---
>
> ### R4. Impact under large-scale and diverse datasets
>
> > “Considering that more and more large-scale and diverse datasets are proposed, … it’s not clear whether the proposed augmentation still provides a large performance gain…”
>
> The new experiments suggest that 3DRot remains useful even with larger or more diverse data and strong default augmentations:
>
> - On **IN10** (SUN RGB-D + ARKitScenes + Hypersim), 3DRot still improves 3D IoU and translation metrics over a strong baseline.
> - On **KITTI**, 3DRot improves moderate Overall AP₃D while being combined with GlobalRotScaleTrans and RandomFlip3D.
> - On **NYU Depth v2**, where the BTS baseline already uses 2D rotations and flips plus color jitter, adding 3DRot further reduces abs-rel and improves δ < 1.25, and the gains carry over to cross-dataset SUN RGB-D.
>
> ---
>
> ### Closing
>
> Thank you again for your helpful suggestions. They led us to substantially broaden the experimental scope.

---

> ### Author Response · Authors · 2025-11-24
> **Comparison of different data augmentations**
>
> In addition, we now include cross-task comparisons of different augmentation families (for monocular 3D cuboid detection, monocular depth estimation, and LiDAR+RGB 3D detection):
>
> **Table 1: Validation results with different augmentations (cuboid detection).**
>
> | Method              | IoU$_{3D}\uparrow$ | Trans (cm)$\downarrow$ | Rot (deg)$\downarrow$ | Size (cm)$\downarrow$ | 5GRAV$\uparrow$ | 10HEAD$\uparrow$ |
> | ------------------- | -------------------- | ---------------------- | --------------------- | --------------------- | --------------- | ---------------- |
> | BASE                | 42.43                | 11.87                  | 23.99                 | 13.09                 | 77.03           | 45.39            |
> | BASE + Color jitter | 41.51                | 12.24                  | 22.48                 | 13.07                 | **78.84**       | 48.16            |
> | BASE + 3DRot        | **44.41**            | **11.13**              | **21.91**             | **12.50**             | 77.26           | **49.06**        |
>
> *Trans and Size are reported in centimeters; Rot is the mean absolute rotation error in degrees. 5GRAV denotes accuracy within $5^\circ$ of the ground-truth gravity axis, and 10HEAD denotes accuracy within $10^\circ$ of the ground-truth heading. All configurations are trained without any flipping augmentation.*
>
> ---
>
> **Table 2: Full BTS ResNet-50 results with all augmentation variants (monocular depth).**
>
> | Family         | Aug.                | NYU abs$\downarrow$ | NYU rmse$\downarrow$ | NYU a1$\uparrow$ | SUN abs$\downarrow$ | SUN rmse$\downarrow$ | SUN a1$\uparrow$ |
> | -------------- | ------------------- | ------------------- | -------------------- | ---------------- | ------------------- | -------------------- | ---------------- |
> | `base`         | plain               | 0.2105              | 0.6765               | 0.6582           | 0.2891              | 0.7587               | 0.5366           |
> | `proj_family`  | flip                | 0.2027              | 0.6828               | 0.6574           | 0.2704              | **0.7408**           | 0.5485           |
> | `proj_family`  | 2DRot               | 0.2059              | 0.6460               | 0.6818           | 0.2835              | 0.7567               | 0.5343           |
> | `proj_family`  | 3DRot$_{2{-}2{-}5}$ | **0.1940**          | **0.6370**           | **0.6954**       | **0.2684**          | 0.7414               | **0.5494**       |
> | `proj_family`  | 3DRot$_{3{-}3{-}5}$ | 0.1965              | 0.6430               | 0.6918           | 0.2733              | 0.7504               | 0.5429           |
> | `color_family` | color               | **0.1879**          | **0.6102**           | **0.7134**       | **0.2532**          | **0.6867**           | **0.5932**       |
>
> *Models are trained on NYU Depth v2 and evaluated on NYU Depth v2 and SUN RGB-D. `base` uses the default BTS schedule (including 2D rotation and horizontal flips. 2DRot, treating depth as a fourth image channel but without enforcing a true roll around the principal point); `proj_family` applies projective augmentations (flip, in-plane 2D rotation baseline, or our camera-centric 3DRot$_{2{-}2{-}5}$/3DRot$_{3{-}3{-}5}$); `color_family` adds stronger photometric jitter.*
>
> ---
>
> **Table 3: Effect of $(G,F,R)$ configurations in MVX-Net ablations on KITTI (LiDAR+RGB 3D detection).**
>
> | $(G,F,R)$ | 3D box    | BEV       | AOS       |
> | --------- | --------- | --------- | --------- |
> | (0,0,0)   | 44.78     | 53.91     | 50.08     |
> | (1,0,0)   | 61.86     | 68.96     | 66.69     |
> | (0,1,0)   | 51.00     | 59.06     | 54.61     |
> | (0,0,1)   | 50.80     | 59.65     | 54.58     |
> | (1,1,0)   | 63.62     | 70.53     | 66.63     |
> | (0,1,1)   | 56.45     | 63.47     | 62.59     |
> | (1,0,1)   | 61.18     | 68.85     | 64.55     |
> | (1,1,1)   | **65.16** | **71.09** | **67.18** |
>
> *$(G,F,R)$ indicates whether GlobalRotScaleTrans (G), RandomFlip3D (F), and AugRotate3D (R) are enabled $(0/1)$. This table summarizes the interaction between these 3D augmentations for MVX-Net on KITTI (moderate AP metrics). GlobalRotScaleTrans is a pure point-cloud augmentation and does not modify the 2D images.*

---

### Author Response · Authors · 2025-12-02
**Author Summary of 3DRot Revisions, New Experiments, and Clarified Positioning**

We briefly summarize our method and the major experimental updates below.

---

## 1. Brief summary & main contribution

3DRot is a plug-and-play augmentation that can update RGB, camera intrinsics, and 3D labels (poses, boxes, depth, LiDAR points) via a single closed-form mapping. It only requires a calibrated pinhole camera and applies the same camera-centric transform to any aligned modality (RGB, depth, LiDAR), so it preserves projective geometry while enriching training data along 3D rotations.

Our goal is to expand the very limited augmentation toolbox available for **RGB-based 3D tasks** and to improve **rotation robustness** under scarce 3D annotations. We now validate 3DRot across three settings: monocular 3D detection, monocular depth estimation, and LiDAR+RGB detection, which are all important for indoor robots and other mobile agents that frequently see tilted or atypical viewpoints.

---

## 2. Pointer to marked changes in the PDF

We updated the PDF with color-marked edits.

---

## 3. Original vs. new tasks and datasets

### Original setting (submitted version)

- **SUN10 (SUN RGB-D 10-class split, DINO-X + Cube R-CNN)**
  AP3D0.5 ↑: 35.70 → 38.11 (+2.41)
  IoU3D ↑: 43.21 → 44.51 (+1.30)
  Rot ↓ (deg): 22.91 → 20.93 (−1.98)

- **IN10 (cross-domain indoor split, DINO-X + Cube R-CNN)**
  IoU3D ↑: 30.39 → 30.59 (+0.20)
  Trans ↓ (cm): 22.97 → 22.75 (−0.22)

### New tasks and datasets added after reviews

- **Monocular depth estimation (NYU Depth v2 → SUN RGB-D, BTS ResNet-50)**
  NYU Depth v2: AbsRel↓ 0.1783 → 0.1685 (≈5.5% relative drop)
  SUN RGB-D: AbsRel↓ 0.2502 → 0.2333 (≈6.8% relative drop)

- **LiDAR+RGB 3D detection (KITTI, MVX-Net)**
  On KITTI detection, adding 3DRot on top of the standard LiDAR+RGB augmentations improves **3D AP↑**:
  Overall 3D AP↑: 63.85 → 65.16 (+1.31)
  Pedestrian 3D AP↑: 55.12 → 59.26 (+4.14)

---

## 4. Original vs. new augmentation comparisons

- **Monocular 3D detection (SUNRGB-D / IN10, DINO-X + Cube R-CNN)**
  In addition to the original comparisons with flips and chirality, we now also compare color jitter vs. 3DRot without any flipping (see Appx. H).

- **Monocular depth (NYU Depth v2 → SUN RGB-D, BTS)**
  We add comparisons between the original BTS schedule and variants with flipping, 2D rotation (treating depth as a fourth image channel without enforcing a true roll around the principal point), color jitter, and 3DRot.

- **LiDAR+RGB detection (KITTI, MVX-Net)**
  In addition, we compare the standard LiDAR+RGB setup against variants with GlobalRotScaleTrans (LiDAR-only augmentation), cross-modal flipping, and their combinations with 3DRot.

---

## 5. Novelty vs prior work and relation to existing augmentations

We focus on **non-generative, non-insertion, plug-and-play augmentations**. In this space, most RGB-based 3D pipelines still train with a very small set of operations:

(i) geometry-consistent crops/resizes with updated intrinsics (but inevitable information loss)

(ii) horizontal flips (and occasionally vertical flips)

(iii) global color jitter

Among them, horizontal flip is by far the most widely use simple, label-preserving transform that is routinely applied cross-modally (the same operation on RGB, depth, and LiDAR). Rotation augmentation around the camera’s optical center is essentially absent from these standard training recipes.

Several works have started to explore **rotation-like** effects, but each makes strong assumptions. [Keypoint-based monocular 3D detection](https://ieeexplore.ieee.org/document/9223480) perturbs roll/pitch/yaw using full RGB-D frames, so it cannot be used when only object poses or LiDAR are available. [Two-level data augmentation](https://arxiv.org/abs/2210.10756) enforces a shared ground plane across calibrated views and is intrinsically coplanar. [GroundMix](https://arxiv.org/abs/2408.11958) reports small in-plane rotations around the optical axis with visually plausible cuboids, but only adjusts yaw and does not provide a general, analytic formulation for arbitrary scenes.

By contrast, 3DRot is a depth-free, optical-center rotation/reflection that rotates the camera in 3D and applies the induced homography to RGB and all 3D labels, extending plug-and-play, geometry-consistent augmentation beyond crops and flips across **RGB-only, RGB–depth, and LiDAR+RGB** pipelines.

---

## 6. Limitations and failure modes

3DRot has practical limitations: its effective rotation range is dataset-dependent (e.g., large roll angles can hurt classes whose camera rigs are nearly level), and when intrinsics are missing or only approximate we must first normalize to a virtual camera or estimate them from 2D–3D correspondences. More broadly, 3DRot is a basic geometry-consistent building block rather than a full remedy for 3D data scarcity, and is meant to complement copy-paste or generative augmentation schemes rather than replace them.

---

### Meta-Review · Area_Chair_CTcG · 2026-01-07

**Summary:**

The reviewers were mainly concerned about the limited novelty of the approach, its incremental nature relative to existing methods, and the modest empirical gains. They also questioned whether the contribution was strong enough for a major conference despite being technically sound.

**Reviewer Concerns:**

The rebuttal addressed most of the reviewers’ concrete and technical concerns. In particular, it substantially strengthened the empirical validation by adding new tasks (monocular depth estimation and LiDAR+RGB detection), new datasets (NYU Depth v2, SUN RGB-D, KITTI), and additional models, directly responding to concerns about limited evaluation scope and generalizability. It also clarified implementation details such as rotation angle ranges, padding and principal point updates, chirality-preserving flips, and the handling of camera intrinsics, as well as improved the clarity and structure of the method and notation.

However, some higher-level concerns remain partially outstanding. In particular, the perception of limited novelty relative to existing augmentation and diffusion-based pipelines is not fully resolved, and while the empirical gains are consistent, they remain moderate. As a result, although the rebuttal successfully addresses most technical weaknesses, it does not fully overturn concerns about the overall level of contribution and impact.

**Reviewer Scores:**

Initially, one reviewer recommended acceptance and three recommended rejection. After the rebuttal, one or two reviewers might slightly increase their scores due to the added experiments and clarifications, but the main concerns about limited novelty would remain, so the overall outcome would not change.

---

### Decision · Program_Chairs · 2026-01-26

Reject